# Critical factors for precise and efficient RNA cleavage by RNase Y in *Staphylococcus aureus*

**Alexandre Le Scornet**[1], **Ambre Jousselin**[1], **Kamila Baumas**[1], **Gergana Kostova**[2], **Sylvain Durand**[2], **Leonora Poljak**[1], **Roland Barriot**[1], **Eve Coutant**[1], **Romain Pigearias**[1], **Gabriel Tejero**[1], **Jonas Lootvoet**[1], **Céline Péllisier**[1†], **Gladys Munoz**[1], **Ciarán Condon**[2], **Peter Redder**[1]*

1 Laboratoire de Microbiologie et Génétique Moléculaires (LMGM), Centre de Biologie Intégrative (CBI), Université de Toulouse, CNRS, Université Toulouse III–Paul Sabatier, Toulouse, France, 2 UMR8261 CNRS, Université Paris Cité, Institut de Biologie Physico-Chimique, Paris, France

† Deceased.

* peter.redder@univ-tlse3.fr

**Data Availability Statement:** The Illumina sequencing data from this study are available in the SRA database, accession number PRJNA1005908.

## Abstract

Cellular processes require precise and specific gene regulation, in which continuous mRNA degradation is a major element. The mRNA degradation mechanisms should be able to degrade a wide range of different RNA substrates with high efficiency, but should at the same time be limited, to avoid killing the cell by elimination of all cellular RNA. RNase Y is a major endoribonuclease found in most Firmicutes, including *Bacillus subtilis* and *Staphylococcus aureus*. However, the molecular interactions that direct RNase Y to cleave the correct RNA molecules at the correct position remain unknown. In this work we have identified transcripts that are homologs in *S. aureus* and *B. subtilis*, and are RNase Y targets in both bacteria. Two such transcript pairs were used as models to show a functional overlap between the *S. aureus* and the *B. subtilis* RNase Y, which highlighted the importance of the nucleotide sequence of the RNA molecule itself in the RNase Y targeting process. Cleavage efficiency is driven by the primary nucleotide sequence immediately downstream of the cleavage site and base-pairing in a secondary structure a few nucleotides downstream. Cleavage positioning is roughly localised by the downstream secondary structure and fine-tuned by the nucleotide immediately upstream of the cleavage. The identified elements were sufficient for RNase Y-dependent cleavage, since the sequence elements from one of the model transcripts were able to convert an exogenous non-target transcript into a target for RNase Y.

## Author summary

In order to correctly regulate the level of RNAs, bacteria require their RNA to be continuously synthesised and degraded. However, even related bacterial species can have different sets of ribonucleases, each with their own target criteria. Here we explore which sequence elements of an RNA are important for being targeted by the endoribonuclese RNase Y in the two bacteria, *Staphylococcus aureus* and *Bacillus subtilis*. We specifically examine the

**Funding:** PR was in part supported by Société Académique de Genève (https://sacad.ch/) and Axes Thématiques Prioritaires de l'Université Paul Sabatier (https://www.univ-tlse3.fr/). SD and CC were supported by funds from the CNRS (UMR8261), Université Paris Cité and the French National Research Agency through ANR-16-CE12-0002 (BaRR) and LABEX DYNAMO (ANR-LABX-011). The funders did not play any role in the study design, data collection and analysis, decision to publish, or preparation of the manuscript.

**Competing interests:** The authors have declared that no competing interests exist.

RNase Y dependent cleavage of two transcripts that have homologs in both bacteria. We identify a short single-stranded regions immediately downstream of the cleavage position, which can be modified to change the cleavage efficiency up to 20-fold. We furthermore discover that a secondary structure a few nucleotides downstream of the cleavage is required for cleavage and that the positioning of the cleavage can be modulated by moving this structure.

## Introduction

RNA turnover is a central element in the gene regulation of all organisms, but especially so in bacteria where mRNA half-lives are typically less than a minute [1–3]. These short RNA life-spans allow the cells to control gene expression at the transcriptome level, since a simple pause in the synthesis of a given mRNA will rapidly lead to a complete depletion of this mRNA. The RNA decay machinery is continuously active to balance out RNA synthesis, but that does not mean that all RNAs are treated equally by the RNA decay machinery. Ribosomal RNA, tRNAs, several small regulatory RNAs, and even a few mRNAs exhibit half-lives that are longer than the generation time of the cells [4–6], strongly suggesting that specific factors protect certain RNAs from degradation or, alternatively, target specific RNAs for degradation. RNases can either degrade an RNA from the 5' or 3' ends (5' and 3' exoribonucleases, respectively) or cleave the RNA internally (endoribonucleases), in which case the endoribonuclease in theory has as many potential cleavage sites as there are phosphodiester bonds in the RNA molecule. Endoribonuclease cleavage events can lead to three different outcomes: i) cleavage leads to degradation of the entire RNA molecule, ii) cleavage leads to two stable RNA molecules, iii) cleavage leads to rapid degradation of either the upstream or downstream fragment, whereas the remaining fragment is more stable. Indeed, there are multiple examples of mRNAs that are cleaved to produce a mature mRNA or regulatory RNA. For example, the mRNA of the *saePQRS* virulence regulation operon in *Staphylococcus aureus* is cleaved by RNase Y between the *saeP* and *saeQ* coding sequences, leading to rapid degradation of the *saeP* fragment and a stable *saeQRS* portion of the mRNA. Consequently, the *saeQRS* mRNA has a much higher steady state level than *saeP* mRNA, even though they are transcribed (and regulated) simulta-neously [7]. Importantly, if such cleavage events maintain the correct homeostasis between the proteins coded by the operons, then there is an evolutionary pressure to conserve both the ele-ments that position the cleavage at the correct nucleotide and the elements that ensure that it happens with the appropriate efficiency. If the positioning is different, then the resulting cleav-age products may have altered stability; if the cleavage efficiency is too high or too low, then the relative abundance of the two cleavage products may be inappropriate, leading to misregu-lation of the gene products.

RNase Y represents one of the three major families of bacterial RNA decay systems (the two others being RNase E and RNase J). RNase Y is ubiquitous in the Firmicute phylum of bacteria and is scattered in most other bacterial phyla, with the notable exceptions of Cyanobacteria, Alpha-, Beta- and Gamma-proteobacteria [8]. Even prior to the discovery of the ribonuclease activity of RNase Y [9], the gene encoding this enzyme (*rny/cvfA*) had been determined to be crucial for *S. aureus* virulence in silkworms [10], a finding that was further confirmed by a mouse study [7]. RNase Y from *S. aureus* and *Bacillus subtilis* are highly similar (68% identity and 78% similarity) and have an N-terminal membrane anchoring alpha-helix, followed by a disordered coiled-coil domain, an RNA binding KH domain, a HD domain containing the active site and a C-terminal region of unknown function (S1 Fig) [11,12]. The membrane

anchor is not essential for enzymatic activity, but deleting it does lead to slower cell division in *S. aureus* [5]. This effect is presumably due to RNase Y activity at inappropriate intracellular locations, since mutants with the RNase Y active-site inactivated, or indeed with the whole RNase Y gene deleted, exhibit growth similar to the wild-type strain, although the virulence is greatly diminished in both mouse and silkworm infection models [5,7,10]. In contrast to the minor effect in *S. aureus*, deletion of RNase Y in *B. subtilis* results in growth defects and severe physiological changes to the bacteria [13]. Deletion of RNase Y in *S. aureus* increases the half-lives of about 15% of the transcripts [5]. This relatively small number indicates that mechanisms are in place to select the RNase Y target RNAs among the global population of RNA in the cell. Furthermore, the RNAs are not cleaved at random positions, but are consistently cleaved at specific sites [5,7,14]. Finally, it has been observed for multiple targets that RNase Y does not cleave all the molecules of a given RNA and that its efficiency is limited by unknown factors.

RNase Y interacts with several other proteins, and different studies sometimes find different interaction partners. In *B. subtilis* there is for example some evidence for interactions with the 3' exonuclease PNPase, whereas this interaction was not identified in *S. aureus* [11,15]. All reports from *S. aureus* and *B. subtilis* agree that both the glycolytic enzyme enolase and the DEAD-box RNA helicase CshA interact with RNase Y [11,12,15–17]. However, these interactions are thought to be transitory, since several attempts to isolate an RNase Y-based degradosome complex have failed. In contrast, RNase Y does appear to form a stable complex with RicT in *B. subtilis*, which is loaded onto RNase Y from the RicA-RicF-RicT complex (also annotated as YmcA-YlbF-YaaT, respectively) [18]. The loading of RicT is required for RNase Y-cleavage of many (but not all) RNA targets in *B. subtilis* and at least one target in *S. aureus* [19].

In an effort to understand the role of RNase Y within the larger context of RNA decay, transcriptomic assays have been performed in *S. aureus*, *B. subtilis* and *Streptococcus pyogenes*, to identify the exact cleavage sites of RNase Y. Briefly, these assays compare RNA ends found in wild-type and RNase Y mutant strains; RNA ends that appear in the wild-type but not in the RNase Y mutant are presumed to be generated by an RNase Y cleavage. This approach has led to the identification of 99, 27 and 190 precise RNase Y cleavage sites in *S. aureus*, *B. subtilis* and *S. pyogenes*, respectively [5,19,20]. Comparisons of the sequences surrounding the RNase Y cleavage positions in *S. aureus* and *S. pyogenes* revealed a strong preference for a purine immediately upstream of the cleaved phosphodiester bond, with a preference for G over A. No additional consensus sequence elements could be identified at the primary sequence level, but *in silico* analyses of the putative secondary structures revealed a tendency to find potential hairpin structures in the vicinity of (but not overlapping with) the cleavage sites [5,20]. These findings were in concordance with data from the *S. aureus sae* operon, where RNase Y cleaves 7 nucleotides upstream of a 21 bp *in silico*-predicted intra-molecular duplex, formed by the two complementary regions. Deletions of either complementary region abolished RNase Y cleavage, as did point mutations in one of the regions that would prevent duplex formation [7,14]. Marincola and Wolz also showed that a 13 bp deletion encompassing the cleavage position generated a new cleavage site that was 13 nt upstream of the native site, and a similar shift of 20 nt was observed with a deletion of 20 bp. This suggested a molecular ruler mechanism, where the cleavage position was chosen based on the distance from a feature downstream, potentially the base of duplex between the complementary regions [14]. Nevertheless, key questions remained unanswered: are the sequence elements that recruit RNase Y identical to those that direct RNase Y to cleave at a specific position, and are the downstream complementary regions, such as those found in the *sae* mRNA, sufficient to cause RNase Y cleavage of other RNA molecules?

In this study we explore the relationship between RNase Y and several different target RNAs, and we establish that the nucleotide sequence surrounding an RNase Y cleavage site is sufficient for cleavage and can convert a non-substrate RNA into a substrate. We also use systematic mutational analyses to determine which nucleotides in the vicinity of the cleavage site are required for efficient cleavage and which nucleotides determine the exact cleavage position. These findings are shown to be general within the Firmicutes, since homologous operons are cleaved by RNase Y in both *S. aureus* and *B. subtilis*, and both RNases and RNA substrates from one bacterium can replace their equivalent in the other. Finally, we show that the *S. aureus gapR* transcript has evolved to maintain a precise cleavage position, but with low cleavage efficiency, and that the efficiency can be modulated by individual point mutations around the cleavage site.

## Results

### Identifying model RNase Y cleavage sites

Endoribonucleolytic cleavage events can either be useful to the cell (increase fitness), detrimental to the cell (decrease fitness), or be neutral (no change in fitness). There is an evolutionary pressure to avoid RNase Y target sequences that are detrimental to the cell, and such RNA sequences are therefore not expected to be found in a strain possessing the enzyme. To distinguish between neutral cleavage sites and beneficial cleavage sites, we reasoned that a useful cleavage site/event would be much more likely to be evolutionary conserved, since a neutral event can be lost by genomic drift without any fitness loss to the cell.

To identify evolutionary conserved cleavage sites, of RNase Y from *S. aureus* and *B. subtilis*, (SaY and BsY, respectively), we first compared genes with known RNase Y cleavage sites in both bacteria [5,19], and identified the transcript-pairs Sa-*gapR*/Bs-*cggR* and Sa-*glnR*/Bs-*glnR* that are conserved in both organisms and cleaved by their respective RNase Ys at similar positions (Fig 1A). The Sa-*gapR*/Bs-*cggR* operons encode a sugar-binding transcriptional regulator, glyceraldehyde-3-phosphate dehydrogenase, enolase and other enzymes of the glycolytic pathway [21,22], whereas the Sa-*glnR*/Bs-*glnR* operon encodes glutamine synthetase repressor and glutamine synthetase. To be able to modify these metabolic transcripts without altering the growth of the cell, we cloned the regions surrounding the RNase Y cleavage sites upstream of the transcriptional terminator in the *E. coli-S. aureus* shuttle-vector pEB01 (Fig 1B). The pSa-Gap construct was designed with an 864 bp (288 codons) deletion within the Sa-*gapR* open reading frame. This in-frame deletion preserved the start and stop codons, but shortened the produced ΔGapR peptide to 51 amino acids, thus ensuring that no functional GapR could be produced from the multi-copy plasmid to interfere with cell physiology, as well as reducing the transcript to a manageable length for cleavage detection by both Northern blotting and EMOTE (Exact Mapping Of Transcriptome Ends, see Materials and Methods). In the same way, the pBsCgg construct was made by deleting 708 bp (236 codons) from the *cggR* open reading frame leaving a short (106-codon) ΔCggR peptide. A similar approach was used to study cleavage of the conserved RNase Y target *glnR*. pSaGln and pBsGln were constructed with 150 bp (50 codons) and 189 bp (63 codons) deletions, respectively, in the *glnR* open reading frames.

The resulting constructs were introduced into *S. aureus* wild-type (WT) and RNase Y deletion mutants (ΔY). RNA was prepared from the transformed *S. aureus* strains, and the cleavage (or lack thereof) was monitored by Northern blotting. We were able to detect the full-length transcripts and the smaller RNase Y cleavage products for all four constructs (Fig 1C) using the probe P1, which hybridises to the vector-derived portion of the mRNA upstream of the transcription terminator (Fig 1B). Lack of downstream cleavage product in the ΔY strain confirmed that the cleavage was indeed RNase Y-dependent.

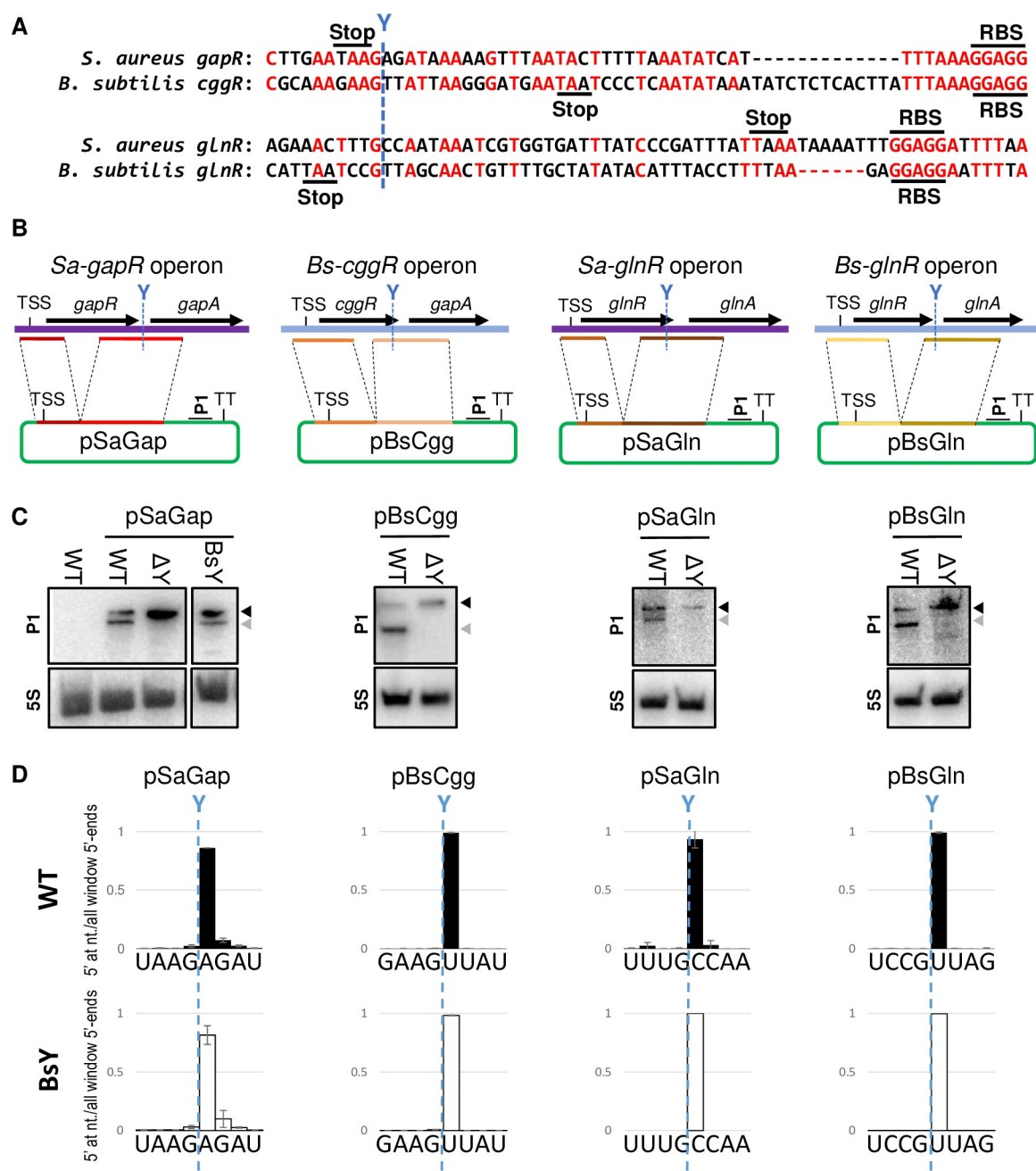

**Fig 1. RNase Y cleaves the model transcripts.** A) Sequence alignment of *S. aureus* and *B. subtilis* transcripts with the published RNase Y cleavage sites indicated [5,19,21]. B) Schematic representation of the *Sa-gapR*, *Sa-glnR*, *Bs-cggR* and *Bs-glnR* operons in *S. aureus* and *B. subtilis*. TSS indicates the transcription start site, and the native RNase Y cleavage sites are indicated by blue dotted lines. Coloured lines show the regions cloned in front of the transcription terminator (TT) of the pEB01 vector (green). The location of the P1 probe is indicated. C) The four constructs from panel B were transformed into the WT and ΔY strains, and RNase Y cleavage was detected by Northern blotting using the P1 probe to detect the full-length (black arrowhead) and the smaller RNase Y cleavage product (grey arrowhead). A probe against 5S rRNA (5S) is used as loading control. pSaGap was also introduced into the *S. aureus* strain BsY, which expresses *B. subtilis* RNase Y from the *S. aureus rny* promoter at the native *S. aureus* chromosomal *rny* locus. D) EMOTE data for cleavage of the four wild-type vector constructs. Columns indicate the proportion of detected RNA molecules with 5'-ends at each position within the shown window (the sum of all columns is 1). WT (black columns): *S. aureus* WT, BsY (white columns): *S. aureus* with SaRNase Y coding sequence substituted with the BsRNase Y coding sequence. EMOTE data from the *S. aureus* ΔY strain is shown in S3A Fig.

It has previously been shown that the upstream cleavage product of the *B. subtilis cggR* operon mRNA is quickly degraded [22], and this was also the case for the equivalent upstream cleavage product from the pSaGap transcript, which we could only detect when using a strain where the 3' exonuclease PNPase was deleted (S2 Fig). The rapid degradation of the upstream cleavage product in a WT strain led us to use the downstream product (and probe P1) as an indicator for cleavage in the rest of this study.

## All necessary information for cleavage selectivity is found within the target RNA nucleotide sequence

The RNase Y proteins from *S. aureus* and *B. subtilis* are highly similar, but not identical (S1 Fig). This difference, or small differences in the RNA substrate sequence, could potentially explain why the conserved RNase Y cleavage positions in *S. aureus* and *B. subtilis* only align roughly. We therefore expressed the relevant region of the *B. subtilis cggR* transcript from a plasmid (pBsCgg) in *S. aureus* and observed that the *cggR* RNA is cleaved by SaY (Fig 1C).

We then used targeted EMOTE to examine the exact cleavage position on the pBsCgg transcript [23]. Briefly, targeted EMOTE is similar to standard 5'-RACE protocols in that an oligo is ligated to the 5'-end of the RNA, followed by reverse transcription with a transcript-specific primer and subsequent PCR amplification. However, instead of cloning the amplified cDNA into a vector, the PCR product is directly sequenced with Illumina sequencing, to determine the exact 5'-nucleotide of more than 5000 individual RNA molecules per RNA sample (in contrast to the 10–20 molecules typically analysed when cloning is used for 5' RACE). This allows for a highly detailed view of the pattern of cleavage positions when the relative frequencies of each 5'-nucleotide are plotted (Fig 1D). In this way we showed that the *B. subtilis cggR* RNA is cleaved at exactly the same position by SaY in *S. aureus* as it is by BsY in *B. subtilis*, and thus that the small difference in cleavage sites is intrinsic to substrate rather than enzyme sequence.

In addition, we replaced the SaY open reading frame on the *S. aureus* chromosome with the BsY open reading frame, and the resulting mutant (ΔSaY::BsY) cleaved both the pBsCgg and pSaGap mRNAs at their respective native sites. These results rule out that a hypothetical organism-specific "RNase Y associated factor" is responsible for guiding the SaY and BsY to their respective sites, since any organism-specific factor would be absent during exogenous expression. This further confirms that the cleavage-site difference between the pSaGap and pBsCgg transcripts is purely due to the differences in nucleotide sequence between the two RNA molecules. Moreover, to ensure that the *Sa-gapR*/*Bs-cggR* pair was not a unique case, we verified that the pSaGln and pBsGln mRNAs were also both cleaved at their native sites, irrespective of the origin of the RNase Y (Fig 1C and 1D).

## *S. aureus* RNase Y is functional in *B. subtilis*

We next wanted to examine whether SaY and BsY were interchangeable in *B. subtilis*. We therefore ectopically expressed SaY and BsY in a *B. subtilis* RNase Y mutant and, after addition of rifampicin to block transcription, examined two well-characterised target RNAs, *glnRA* and *atpIBE*, by Northern blotting. Full-length transcripts were stabilized in the absence of RNase Y (half-lives of <2.5 min *vs.* 14 min for *glnRA* and <2.5 min *vs.* 9.6 min for *atpIBE*) and the processed species (*glnA* and *atpBE*) were no longer detected. When either SaY or BsY was expressed ectopically in *Δrny* cells, then the full-length mRNA was rapidly degraded as observed for the wild-type *B. subtilis* (half-life of <2.5 min for both *glnRA* and *atpIBE*) and the processed forms (*glnA* and *atpBE*) reappeared (Fig 2). Thus, similar to what we observed in *S. aureus*, the enzymes can complement each other in *B. subtilis*.

## Translation is not required for RNase Y cleavage of the pSaGap transcript

The *Bs-cggR* and *Sa-glnR* transcripts are cleaved within an ORF shortly before a stop codon and *gapR* and *Bs-glnR*, are cleaved just after a stop codon but within the ~15 nt region covered by the terminating ribosome (Fig 1A). It was therefore possible that the translational machinery influenced RNase Y target site selection. We therefore constructed two variants of pSaGap that avoid nucleotide changes in close vicinity to the cleavage site but nevertheless modify ribosome positioning/occupancy: pSaGap[NoStart] with the *gapR* start codon mutated, and pSaGap[Δ128] with a 128 nt deletion which causes the ribosome to change reading frame and encounter a stop codon five nucleotides upstream of the original *gapR* stop codon. Northern blot analyses of transcripts from both vectors revealed a prominent cleaved product (S4A Fig) and the pSaGap[Δ128] transcript was further analysed by EMOTE, which confirmed that the cleavage still occurs precisely at the native position (S4B Fig). We therefore conclude that translation does not influence the RNase Y cleavage site position of the pSaGap transcript.

## The upstream G reduces cleavage position heterogeneity

We originally detected a preference for a guanosine residue immediately upstream of the RNase Y cleavage sites in *S. aureus*, and this preference has since been observed for RNase Y

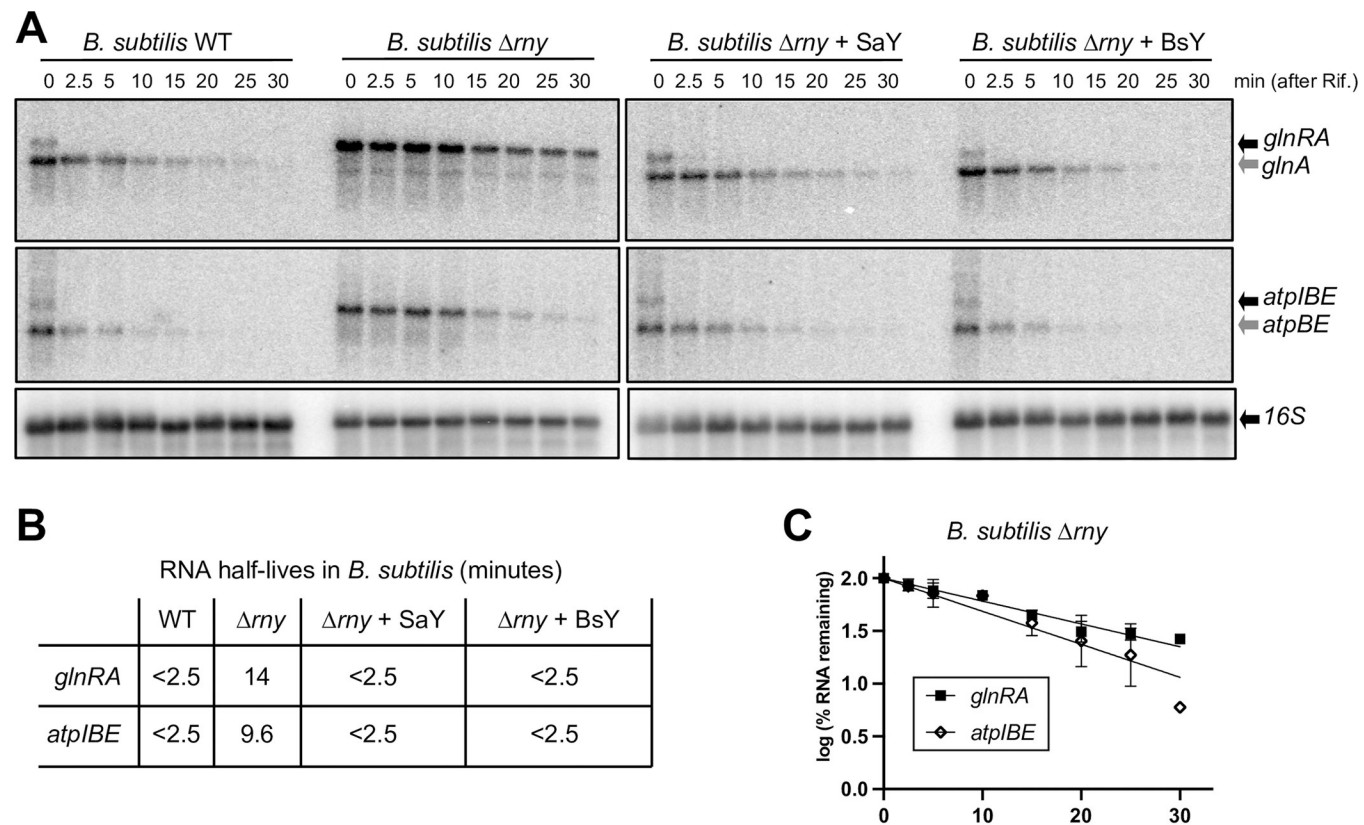

**Fig 2. *S. aureus* RNase Y complements a *B. subtilis* Δ*rny* strain.** A). Northern blots of total *B. subtilis* RNA, showing the disappearance of the uncleaved transcripts (*glnRA* and *atpIBE*, black arrows) after rifampicin addition, accompanied by the appearance of the downstream cleavage products (*glnA* and *atpBE*, grey arrows). "WT": *B. subtilis* wild-type strain. "Δ*rny*": *B. subtilis* with the RNase Y gene deleted. "Δ*rny* + SaY": Δ*rny* strain with *S. aureus* RNase Y expressed from the *amyE* locus. "Δ*rny* + BsY": Δ*rny* strain with *B. subtilis* RNase Y expressed from the *amyE* locus. 16S rRNA was used as loading control. B). Half-lives of the two full-length transcripts (*glnRA* and *atpIBE*) in the four *B. subtilis* strains. *B. subtilis* Δ*rny* was the only strain where the half-lives were long enough to be measured accurately. C). RNA decay curves of *glnRA* and *atpIBE* transcripts in the *B. subtilis* Δ*rny* strain, showing the percentage of remaining RNA at the different time-points.

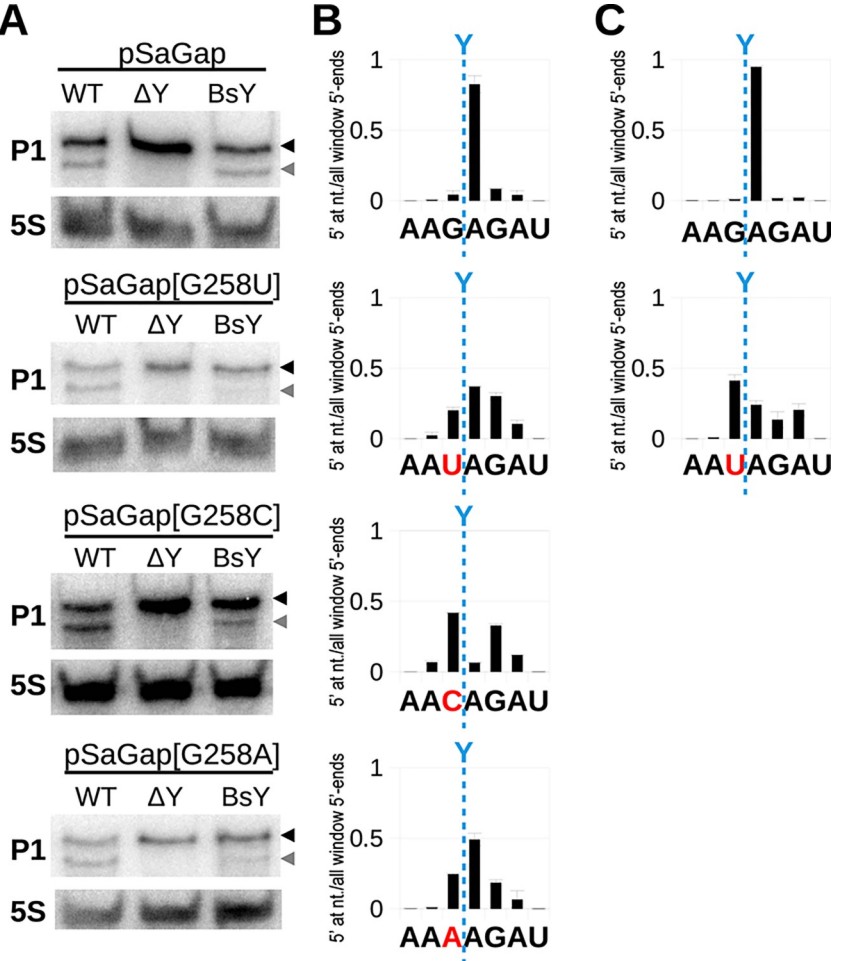

**Fig 3. The upstream G promotes precise cleavage.** A) Northern blots using probe P1 on transcripts from wild-type pSaGap, and versions where the G upstream of the cleavage site has been mutated to U, C and A (pSaGap[G258U], pSaGap[G258C] and pSaGap[G258A]), respectively. Each plasmid was introduced into the WT, ΔY and BsY strains. The full-length transcript and the downstream cleavage product are indicated by black and grey arrowheads, respectively. A 5S rRNA probe was used as loading control. B) EMOTE analysis of the exact cleavage position for *S. aureus* RNase Y. The sequence below each graph shows the nucleotides surrounding the natively preferred cleavage site (blue dotted line). The mutated nucleotide for each plasmid is shown in red. Columns indicate the proportion of detected RNA molecules with 5'-ends at each position within the shown window (the sum of all columns is 1). Two biological replicates were analysed, and the error-bars indicate the standard deviation. C) Same as panel B, but where *B. subtilis* RNase Y has replaced *S. aureus* RNase Y at the native *rny* locus.

from *Streptococcus pyogenes* as well [5,20,24]. To gauge the importance of this nucleotide in RNase Y site selectivity, we generated constructs pSaGap[G258U], pSaGap[G258C] and pSa-Gap[G258A], where this nucleotide was mutated to uridine, cytosine and adenosine, respectively. Northern blots performed on these strains revealed RNase Y-dependent cleavage for each of the three mutants (Fig 3A). To determine whether the RNase Y cleavages were at the exact same sites as in the WT pSaGap transcript, we complemented the Northern blot data with EMOTE targeted to the transcripts expressed from the vector [23].

The EMOTE data revealed that RNase Y cleavage of the pSaGap[G258C] transcript preferentially occurred immediately upstream or downstream of the native cleavage site, whereas in pSaGap[G258A] and pSaGap[G258U], cleavage was still preferentially at the native site, but frequently occurred a few nucleotides upstream or downstream (Fig 3B). In the ΔSaY::BsY

strain the *B. subtilis* RNase Y appears to cleave slightly less efficiently when the G is substituted (Fig 3A), but the cleavage position pattern is the same as for the *S. aureus* RNase Y, with cleavage exactly after the G and somewhat imprecise when the G is substituted by a U (Fig 3C). These results confirmed that the G is not required for cleavage *per se*, but that it plays an important role in avoiding heterogeneity of the RNase Y cleavage position.

## Four sectors are required for efficient RNase Y cleavage of the *gapR* transcript

Previous studies have indicated that RNA secondary structures are important for RNase Y cleavage [14,21], and we therefore used the mFold software [25] to predict the secondary structures surrounding the RNase Y cleavage site. We specifically examined the uninterrupted region from the in-frame deletion in *gapR* to within *gapA* (light red line in Fig 1B), corresponding to position +117 to +504 in our model transcript. Based on this prediction, we divided the region around the RNase Y cleavage site in the pSaGap transcript into six sectors, named I to VI (Fig 4A). We then examined which sectors of the pSaGap transcript were required for the RNase Y cleavage.

Deletion of the large sector I, ending 14 nt upstream of the cleavage site, did not measurably alter the cleavage (Fig 4). We therefore went on to generate a construct where both sectors I and II were deleted (pSaGap[ΔIΔII]), and observed that this abolished cleavage of the resulting transcript (S5A Fig). Here it should be noted that the deletion of both sectors I and II will place a very weak predicted hairpin structure close to the cleavage site (S5B Fig), which theoretically could be the factor that blocks cleavage. Since sector II includes the G immediately upstream of the RNase Y cleavage site, it was also possible that the absence of this G prevented cleavage. We therefore constructed pSaGap[ΔIΔII+G], which was identical to pSaGap[ΔIΔII], except that the G was left intact. However, reintroducing the G was not sufficient to re-establish cleavage (S5A Fig), and no cleavage product could be detected by Northern blotting of pSaGap[ΔII+G] transcript (Fig 4B), although careful analysis of the corresponding EMOTE data did reveal RNA molecules cleaved at the position of the native RNase Y cleavage site, albeit at a level close to the background noise (Fig 4C). The role of sector II was further explored by generating four mutated versions of sector II, all of which retained the G residue: SecIImutant1, where the 12 nucleotides from the hairpin of sector I replaced the 12 nucleotides of sector II; SecIImutant2, where sector II was replaced by the 12 equivalent nucleotides from the SaGln transcript; SecIImutant3, where the first half of sector II was mutated and SecIImutant4, where the second half of sector II was mutated (S5C Fig). All four of these mutant transcripts were cleaved in an RNase Y-dependent manner (S5D Fig), suggesting that sector II must be present for cleavage to occur, but that its sequence is of little to no importance. To determine whether a similar deletion would have the same effect on other transcripts, we deleted the 12 nucleotides upstream of the conserved G residue of pBsCgg and pBsGln (generating pBsCgg[Δ12] and pBsGln[Δ12], respectively), and observed that both of the mutant transcripts were cleaved by RNase Y, although the efficiency of cleavage appears to be lower in pBsGln[Δ12] (S5E and S5F Fig).

A deletion analysis was also conducted at the 3' end of the pSaGap construct, where sector VI was deleted, either alone (pSaGap[ΔVI], Fig 4B), or in combination with sector I (pSaGap[ΔIΔVI], Fig 4D), and neither of these deletions prevented RNase Y cleavage (Fig 4B and 4D). In contrast, when both sectors V and VI were deleted (pSaGap[ΔVΔVI]), then cleavage was undetected by Northern blotting (Fig 4B), although still detectable by EMOTE (Fig 4C). This result suggests that other sectors are partially redundant with sectors V and VI, but that sector V is a key element in cleavage efficiency. However, to our surprise a cleavage product was

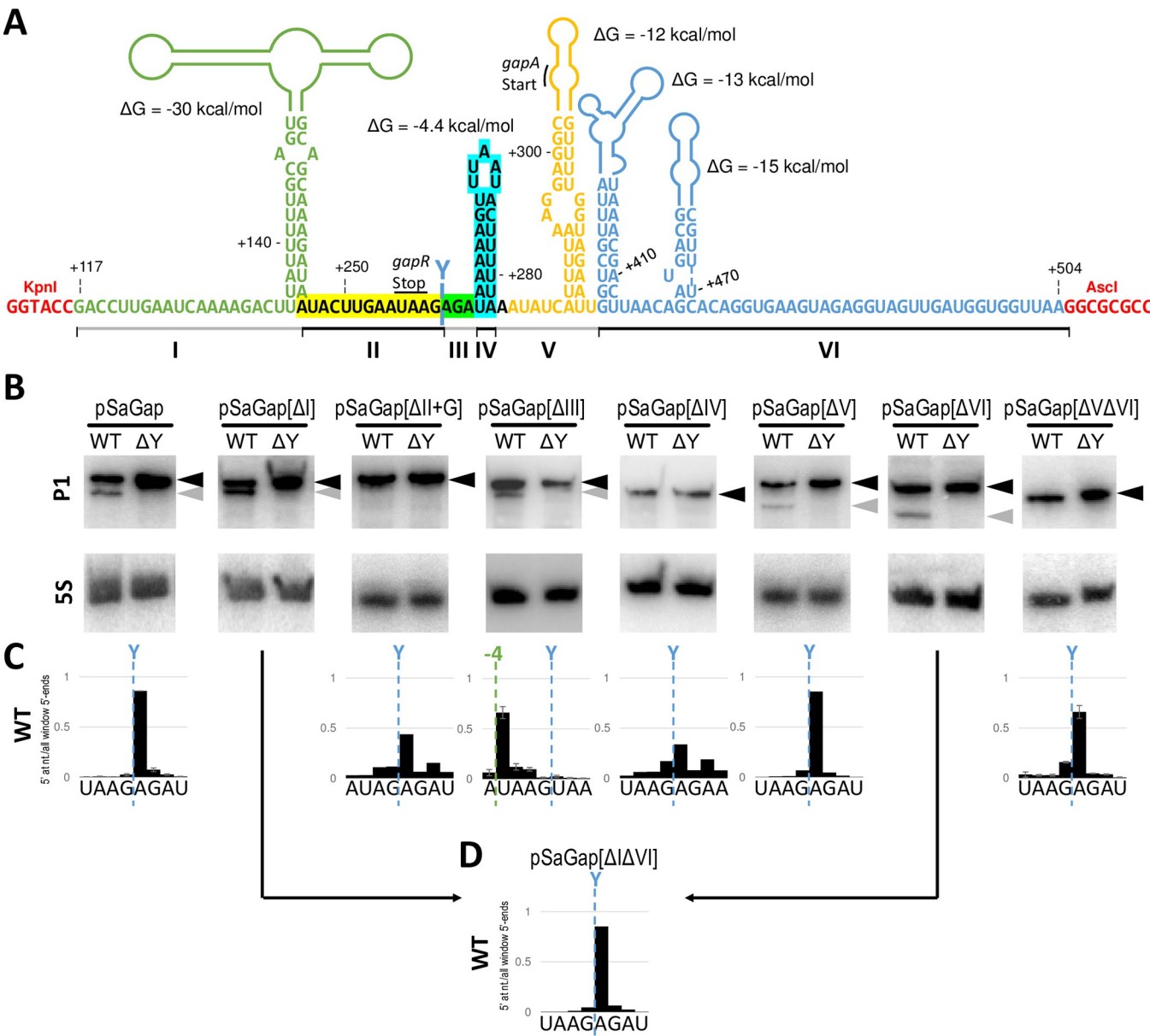

**Fig 4. Deleting sectors of the pSaGap transcript in the region surrounding the RNase Y cleavage.** A) Predicted secondary structures of positions +117 to +504 in the pSaGap vector transcript. We divided the region into six sectors (I-VI), shown in different colours and with black and grey lines below the nucleotide sequence. The native RNase Y cleavage site (Y-cut) is located between sector II and sector III, and the positions of the stop codon of *gapR* and the start codon of *gapA* are indicated. Sectors II, III and IV (marked with yellow, green and cyan background, respectively) are highly important for RNase Y cleavage. Sector I (light green) can be deleted without affecting RNase Y cleavage. Only one of sectors V and VI (orange and blue, respectively) can be deleted without abolishing cleavage. The KpnI and AcsI restriction sites used for cloning the +177 to +504 *gap* operon fragment are shown in red. The predicted minimum free energy for each secondary structure is indicated, but it should be noted that translation will continuously disrupt the secondary structures in sectors I, V and VI. B) Northern blots of the pSaGap transcripts deleted of the various sectors. Black arrowheads denote full-length transcripts and grey arrowheads denote the downstream cleavage products. 5S rRNA was used as loading control. C) EMOTE data showing the exact location of 5' ends in the strains used for Northern blotting in panel B. The green "-4" indicates the 4 nt upstream shift of the cleavage position in the pSaGap[ΔIII] transcript. EMOTE data is presented as number of detected molecules with a given 5'-end divided by the total number of detected molecules within the chosen window. EMOTE data from the ΔY strain can be found in S3B Fig. Arrows indicate that the EMOTE data was obtained for the vector where both sector I and sector VI were deleted (see panel D). D) EMOTE data showing the exact location of 5' ends in the strain carrying the vector where both sector I and sector VI were deleted (pSaGap[ΔIΔVI]), EMOTE data from the ΔY strain can be found in S3C Fig.

visible by Northern blot for a transcript with only sector V deleted (pSaGap[ΔV]) (Fig 4B), and EMOTE confirmed that the cleavage of the pSaGap[ΔV] transcript was at the native position (Fig 4C).

Sector IV is predicted to form a very weak (ΔG of -4.4 kcal/mol) hairpin three nucleotides downstream of the cleavage site (Fig 4A). A cleaved pSaGap transcript was no longer detectable by Northern blotting when sector IV was deleted (pSaGap[ΔIV]). However, similar to our observations for pSaGap[ΔII+G], the corresponding EMOTE data did reveal RNA molecules cleaved at the position of the native RNase Y cleavage site, albeit at a level close to the background noise (Fig 4C), suggesting partial redundancy with another sector. Note that the EMOTE data is presented as the number of detected molecules with a given 5'-end divided by the total number of detected molecules within the chosen window. The vertical axis therefore does not reflect the total intracellular concentration of molecules with a given 5'-end, but rather its abundance relative to the other molecules in the chosen window.

## Sectors II to V are sufficient for RNase Y cleavage of an exogenous transcript

Overall, our deletion analysis suggested that sectors II to IV together with either sector V or VI would be sufficient to define an RNase Y cleavage site. We next wanted to use this knowledge to generate an RNase Y cleavage site in a previously uncleaved transcript. We generated a vector which expressed the heterologous *E. coli fliM* transcript (encoding a flagellar motor protein) with and without the 103 nts of sector II, III, IV and V (Fig 5A). We could detect the full-length *fliM* transcript in both *S. aureus* WT and ΔY strains, as a single uncut RNA by Northern blot (Fig 5B). When the chimeric *fliM*::II-V transcript from the WT strain was examined we could observe both the full-length transcript and a cleavage product of the expected size. However, we were equally able to observe the cleaved *fliM*::II-V product in the ΔY strain, suggesting that the cleavage was not RNase Y dependent (Fig 5C). To understand this unexpected result, we examined the *fliM*::II-V RNA by EMOTE, and discovered two different cleavage sites in the WT strain, only 2 nucleotides apart (Fig 5D). One of these cleavages was unexpectedly detected in both WT and ΔY and is responsible for the cleaved product observed by Northern blot for *fliM*::II-V in the ΔY strain. This means that the insertion of sectors II to V into the *fliM* sequence generated a cleavage site for an as yet unidentified RNase. The other cleavage was located at the exact position where RNase Y normally cleaves the *Sa-gapR* transcript, and this cleavage was dependent on RNase Y, since it disappeared in the ΔY strain (Fig 5D). Thus, the addition of sectors II-V was sufficient to generate an RNase Y-dependent cleavage site in an exogenous RNA.

## Altering the RNase Y cleavage site selection

Sector III consists of the three nucleotides which separate the RNase Y cleavage site from the predicted hairpin of sector IV. Deleting sector III (pSaGap[ΔIII]) did not substantially alter the size or intensity of the cleaved fragment in a Northern blot, but EMOTE revealed that the cleavage site had shifted four nucleotides upstream from the native site (Fig 4C). This suggests that downstream of a cleavage site there is a reference point for measuring the cleavage position. This reference point must be somewhere within sector IV, since removal of sector V from the pSaGap did not change the cleavage position (Fig 4).

## Experimental analysis of secondary structure in sectors II, III and IV

The predicted free energy of the sector IV hairpin is very low (ΔG = -4.4 kcal/mol) and we therefore wanted to test whether individual nucleotides were base-paired or not. For this we

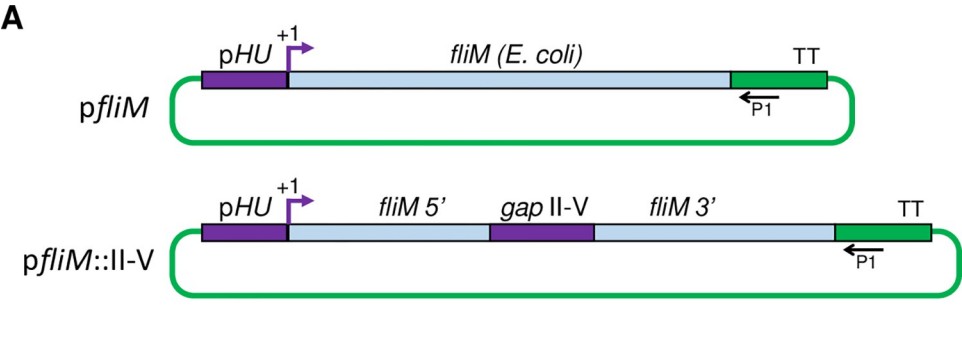

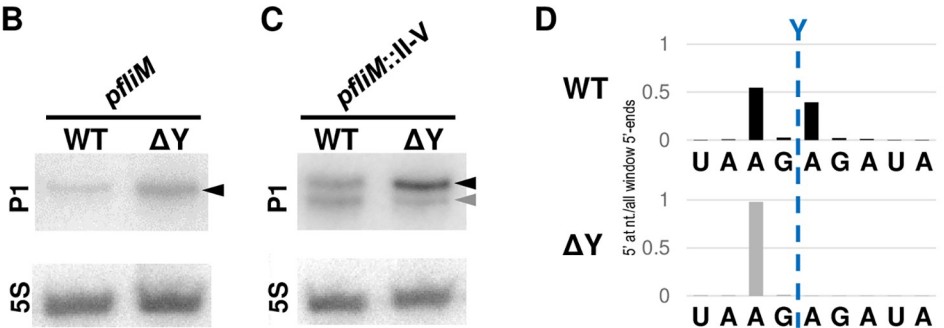

**Fig 5. Generating an RNase Y cleavage site in an exogenous transcript.** A) Overview of the two vectors p*fliM* and p*fliM*::II-V, where regions from *S. aureus* are in purple, regions from *E. coli* are in light blue and regions in green are from the vector backbone. TT indicates the transcription terminator and P1 shows the location where the P1 probe hybridises. *gap*II-V indicates the 103 bp from sector II to V in the *Sa-gapR* operon. B) Northern blot of the *fliM* transcript. The full-length uncleaved RNA is indicated by a black arrowhead. 5S rRNA was used as loading control. C) Northern blot of the *fliM*::II-V transcript, where the uncleaved RNA and downstream cleavage products are indicated by black and grey arrowheads, respectively. 5S rRNA was used as loading control. D) EMOTE data for WT and ΔY strains carrying the p*fliM*::II-V vector. The data from the WT strain (black) shows two distinct peaks at the native RNase Y cleavage site (Y) as well as two nucleotides upstream, whereas the data from the ΔY strain (grey) only shows a single peak two nucleotides upstream of the native RNase Y site. EMOTE data is presented as number of detected molecules with a given 5'-end divided by the total number of detected molecules within the chosen window.

used DMS-MaPseq [26], where dimethyl sulfate (DMS) is first used to methylate non-duplexed A and C nucleotides within the RNA. cDNA is then generated with a special reverse transcriptase (TGIRTIII) which has a very high error-rate when encountering methylated nucleotides. The pool of cDNA is then Illumina-sequenced, and the error-rate of each sequenced nucleotide in the DMS-treated sample can be compared with the error-rate in a mock-treated sample. A proportionally low error-rate for a given nucleotide means that it is protected from methylation by being base-paired.

A and C nucleotides in sectors II and III were reverse-transcribed with high error-rates, meaning that these sectors are single stranded (Fig 6A and 6B). However, the adenosines in the upstream "arm" and the single cytosine in the downstream "arm" of the predicted hairpin in sector IV were all reverse-transcribed with low error rates, meaning that these nucleotides are much more likely to be base-paired (Fig 6A and 6B). In contrast, the adenosines in the predicted hairpin loop exhibited high error-rates, and the same was surprisingly the case for the adenosines at the beginning and at the end of the predicted hairpin stem (A275 and A282) (Fig 6A and 6B). We interpret this to mean that the sector IV hairpin forms on a majority of RNA molecules, but that the hairpin stem "breathes", with base-pairing at both ends continuously disassociating and reforming.

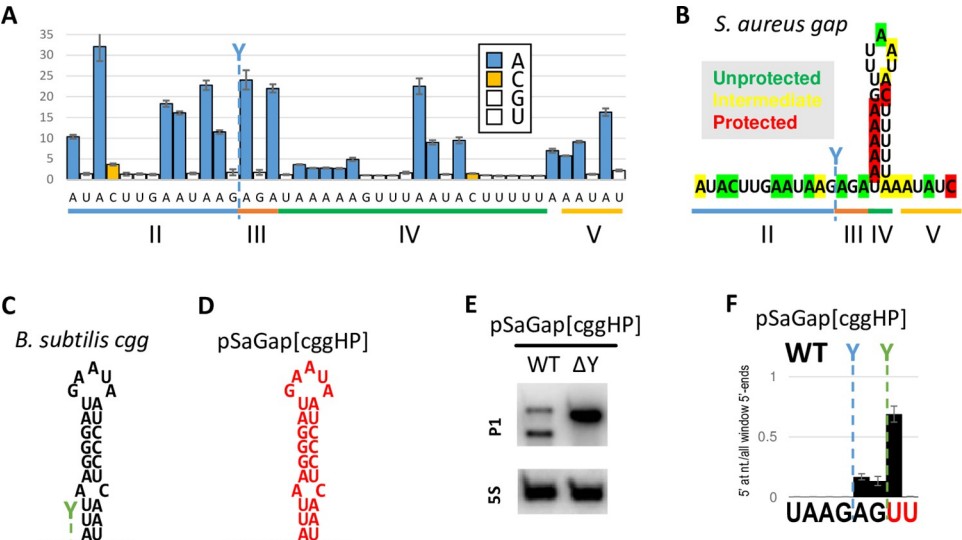

**Fig 6. RNase Y cleavage position depends on hairpin composition.** A) Reverse transcription error rates, normalised with error rates from mock-treated RNA samples, for individual nucleotides around the cleavage site in the *gap* transcript (blue Y). G and U are not affected by DMS and have an average normalised error rate of 1.29 (white columns). A and C (blue and orange columns, respectively) have increased error rates when unprotected by duplex formation. The sectors are indicated below the graph. Standard deviation bars come from three different RNA preparation. B) The predicted *S. aureus gapR* hairpin structure immediately downstream of the cleavage site, correlated with the data from panel A. Protected and unprotected bases are shown in red and green, respectively. Yellow indicates an intermediate signal for bases that presumably are protected in some RNA molecules but not in others. C) *B. subtilis Bs-cggR* hairpin structure immediately downstream of the cleavage site, with the RNase Y cleavage position shown in green. D) The hairpin structure immediately downstream of the cleavage site in the pSaGap construct was exchanged with the hairpin from *Bs-cggR* to obtain pSaGap[ΔIV::cggHP]. Mutated nucleotides are shown in red. E) Northern blot to detect the RNase Y-dependent cleavage of the pSaGap[ΔIV::cggHP] transcript. F) EMOTE signal from the pSaGap[ΔIV::cggHP] transcript. The native cleavage positions of pSaGap and pBsCgg are shown in blue and green, respectively. Mutated nucleotides are in red. EMOTE data is presented as number of detected molecules with a given 5'-end divided by the total number of detected molecules within the chosen window. EMOTE data from the ΔY strain can be found in S3D Fig.

## Exchanging the predicted sector IV hairpins from *Bs-cggR* and *Sa-gapR* alters the cleavage pattern

Similar to what we observe for the *Sa-gapR* transcript, the *Bs-cggR* RNA is also predicted to form a hairpin structure immediately downstream of the cleavage site (Fig 6C), but with the key difference that the *Bs-cggR* cleavage site is only a single nucleotide upstream of the predicted hairpin. We exploited this minor difference in cleavage position, and examined cleavage in a construct where sector IV from pSaGap was exchanged with the corresponding hairpin from *Bs-cggR* (Fig 6D; pSaGap[ΔIV::cggHP]). The hypothesis was that if the position of the base of the hairpin was the defining factor for RNase Y cleavage site selection, then the transcript from pSaGap[ΔIV::cggHP] would be cleaved at the same site as for pSaGap. Alternatively, if the defining factor was not the transition from single-stranded RNA to double stranded RNA, but instead further within the hairpin, then the cleavage site should be identical to that from the pBsCgg transcript. Northern blotting showed that the pSaGap[ΔIV::cggHP] transcript was cleaved efficiently, demonstrating that an exogenous hairpin (from *B. subtilis*) was able to replace the factors in sector IV that are required for efficient cleavage (Fig 6E). EMOTE data furthermore revealed that the major cleavage position was shifted downstream by two nucleotides to coincide with the native cleavage position for the pBsCgg transcript (Fig 6C and 6F). Weak residual cleavage was observed at the native pSaGap position and the

intermediate position of the pSaGap[ΔIV::cggHP] transcript (Fig 6F), possibly influenced by the start of the hairpin duplex or the composition of one of the other sectors. However, the hairpin composition appears to be one of the major factors in cleavage positioning.

## Role of the putative downstream hairpin

In addition to the two predicted hairpins detected in *Sa-gapR* and *Bs-cggR*, hairpins were also predicted to form in Sa-*glnR* and Bs-*glnR*, two and three nucleotides downstream of the native cleavage site respectively. The hairpins are all AU rich, but with one or a cluster of G-C base-pairs close to the loop (Fig 7Ai, 7Bi and 7Ci). Although not all RNase Y cleavage sites are followed by hairpin structures [5], in light of the shifts in cleavage position observed for pSaGap [ΔIII] and pSaGap[ΔIV::cggHP] (green lines in Figs 4C and 6F, respectively), we nevertheless decided to examine whether modification of these predicted hairpins would alter RNase Y cleavage. We first examined whether the sequence of the loop had an impact on cleavage. The five-nucleotide UUAAU loop sequence was inverted to AAUUA (pSaGap[InvLoop], S6B Fig), to alter the sequence while maintaining the minimal potential for additional duplex formation within the transcript. This change in the loop sequence did not prevent RNase Y from cleaving the transcript (S6C Fig), and we therefore focused on the hairpin stem.

## Hairpin G-C base-pairing promotes cleavage

For pSaGap, pBsCgg and pSaGln, the G-C pairing(s) in the stem of the potential hairpin were systematically mutated to destabilise the secondary structure, and Northern blotting showed impaired RNase Y cleavage of these transcripts (Fig 7, panels Aii, Aiv, Bii and Cii). In contrast, mutations restoring the integrity of the putative hairpin with complementary mutations all restored cleavage (Fig 7, panels Aiii, Biii and Ciii). Importantly, faint bands corresponding in size to the cleaved product could be detected in the strains with pSaGap[G268C], pSaGap [C276G] and pBsCgg[GtoC] constructs (Fig 7, panels Aii, Aiv and Bii), indicating that while the G-C base-pairings are important for cleavage of the pSaGap and pBsCgg transcripts, there are other element(s) that can permit RNase Y cleavage.

## The composition of the hairpin stem affects both cleavage efficiency and site selection

Since the previous experiment clearly demonstrated that the G-C base-pair in pSaGap is a key factor in RNase Y cleavage, we generated mutants where the duplex of the stem was either fully inverted or where all but the G-C pair was inverted (Fig 8A). These transcripts are still predicted to form hairpin structures with identical base-pair composition, but RNase Y cleavage was almost undetectable by Northern blotting (Fig 8C). An EMOTE assay revealed that the cleavage position was frequently shifted two nucleotides downstream (Fig 8B). A similar two-nucleotide shift could also be observed when a single U on the downstream "arm" of the stem was mutated into an A (pSaGap[U280A]). In contrast, neither the mutation of the opposing A to a U (pSaGap[A264U]) nor the double mutation to regain base-pairing (pSaGap[A264U, U280A]), gave a strong shift in cleavage site positioning (Fig 8B).

## Mutations immediately downstream of the *Sa-gapR* cleavage site modulate the cleavage efficiency

Having previously observed that deleting the three nucleotides immediately downstream of the cleavage site (i.e. sector III) results in an upstream shift (Fig 4), we wondered whether the three nucleotides of sector III influences the cleavage efficiency. We therefore generated a

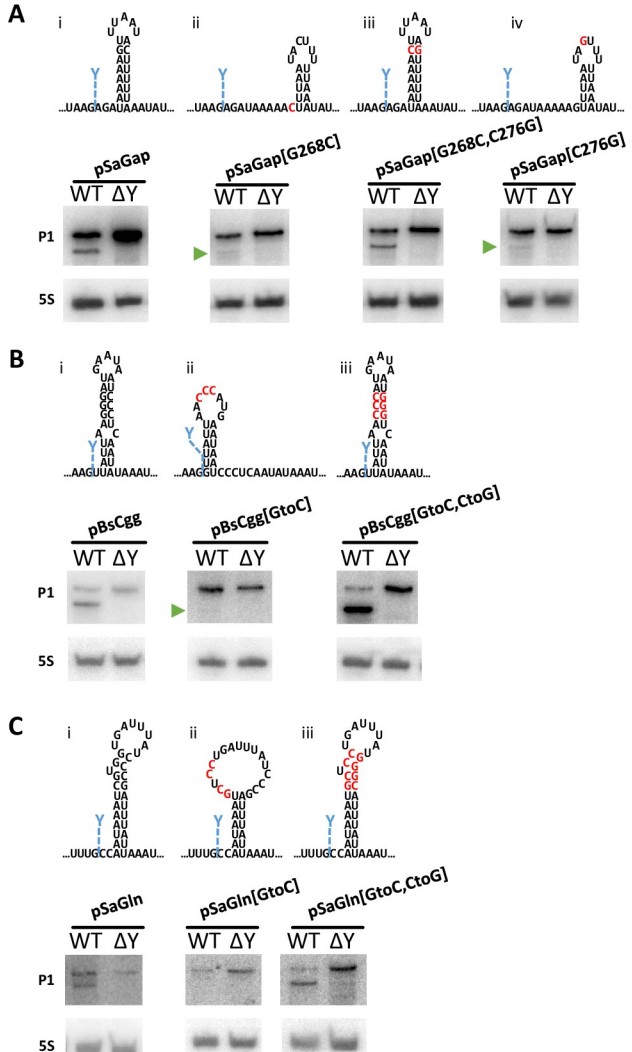

**Fig 7. Hairpin G-C base-pairing promotes cleavage in pSaGap, pBsCgg and pSaGln transcripts.** A). Putative secondary structures of the pSaGap sector IV hairpin with mutated G-C base-pair (mutated nucleotides in red). Northern blots revealed efficient cleavage when the G-C pair was inverted (pSaGap[G268C,C276G]) but only weak cleavage when either the G or the C were mutated (green arrowheads). EMOTE data for these constructs can be found in S7 Fig. B). Putative secondary structures of the pBsCgg downstream hairpin with mutated G-C base-pairs (mutated nucleotides in red). Northern blots revealed efficient cleavage by *S. aureus* RNase Y when the G-C pairs were inverted (pBsCgg[GtoC,CtoG]) but only weak cleavage when the three Gs were mutated to Cs (green arrowhead). C). Putative secondary structures of the pSaGln downstream hairpin with mutated G-C base-pairs (mutated nucleotides in red). Northern blots revealed efficient cleavage when the G-C pairs were inverted (pSaGln[GtoC,CtoG]) but only weak cleavage when CGUGG was mutated to GCUCC (pSaGln[GtoC]). EMOTE data for these constructs can be found in S7 Fig.

library of pSaGap plasmids expressing the 64 possible nucleotide combinations that can be generated within sector III (Fig 9A). The library was transformed into the *S. aureus* wild-type strain, and the relative abundance of cleaved RNA for each of the trinucleotide combinations was measured with EMOTE. The EMOTE signal was normalised by the relative abundance of each of the 64 different plasmids in our library (we assumed that mutation of three nucleotides in the middle of the pSaGap transcript would not significantly modify transcription rates). The wild-type trinucleotide sequence (AGA) turned out to be one of the worst sequences for

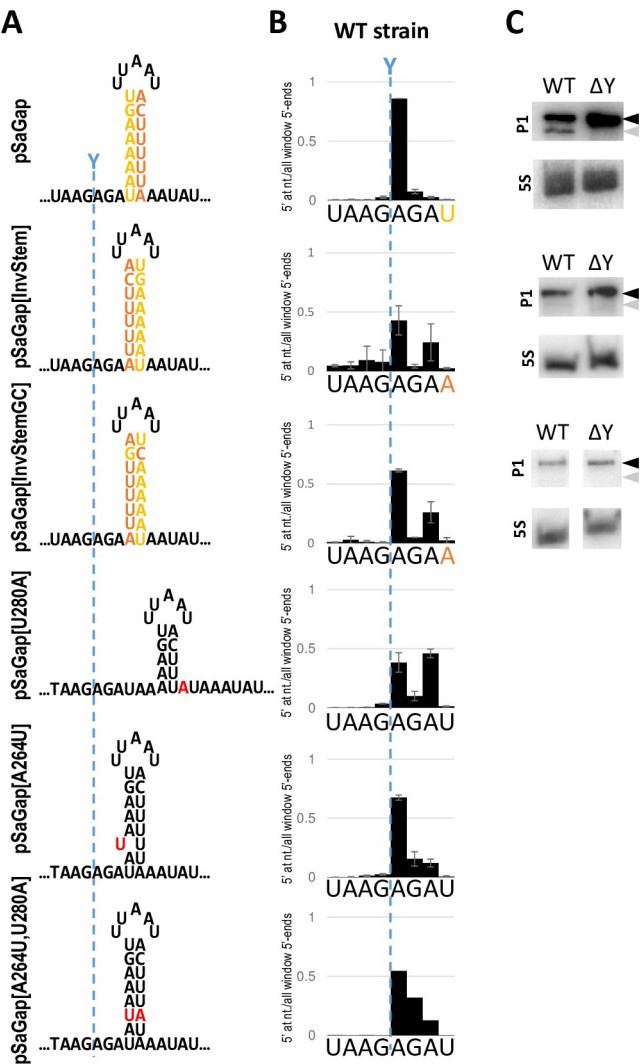

**Fig 8. Modifications of the putative hairpin stem in sector IV.** A) Putative secondary structures of sector IV and derivatives. pSaGap: shown with the upstream and downstream hairpin "arms" in yellow and orange, respectively. pSaGap[InvStem]: the "arms" of the hairpin inverted. pSaGap[InvStemGC]: the "arms" of the hairpin, excluding the G-C base-pair, inverted. pSaGap[U280A]: the third nucleotide in the downstream "arm" mutated to an A, disrupting the potential base-pairing with A264 of the upstream "arm". pSaGap[A264U]: third nucleotide of the upstream "arm" mutated to U, disrupting the potential base-pairing with U280 of the downstream "arm". pSaGap[A264U,U280A]: Both A264 and U280 are mutated as above, resulting in a reconstituted putative base-pairing. The native RNase Y cleavage position is indicated with a blue dotted line. B) EMOTE cleavage profiles of the mutants illustrated in panel A. EMOTE data is presented as number of detected molecules with a given 5'-end divided by the total number of detected molecules within the chosen window. The native RNase Y cleavage position is indicated with a blue dotted line. EMOTE data from the ΔY strain can be found in S3E Fig C) Northern blots of pSaGap, pSaGap[InvStem] and pSaGap[InvStemGC]. Black and grey arrowheads indicate the positions of full-length transcripts and cleavage product, respectively. The cleavage products of pSaGap[InvStem] and pSaGap[InvStemGC] are barely detectable.

RNase Y cleavage efficiency, and transcripts with other NGA trinucleotides are also poorly cleaved (Fig 9B). The cleavage efficiency of transcripts with GNN was also generally low, although this could be due to biases in the detection method (see discussion). To support the EMOTE data with an alternative method, we generated individual pSaGap variants with low efficiency cleavage (pSaGap[IIIGGA] and pSaGap[IIICGA]), very low cleavage efficiency (pSaGap[IIIAUU]) and high cleavage efficiency (pSaGap[IIICUA]). The cleavage of these

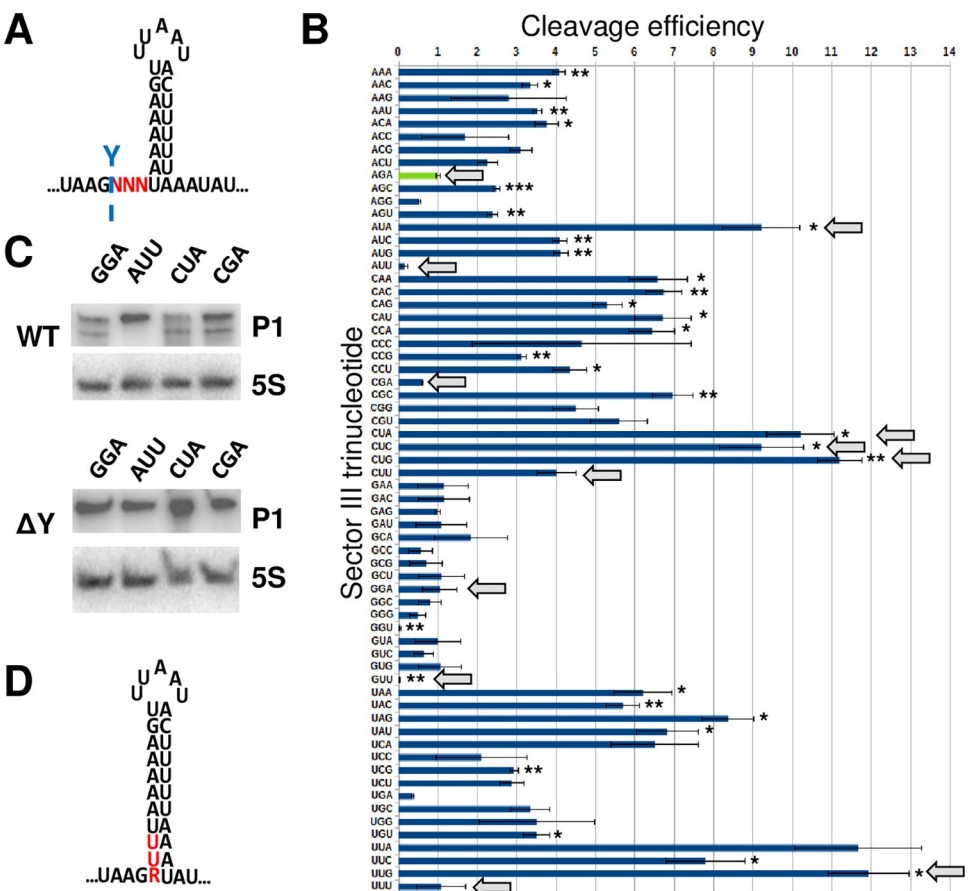

**Fig 9. The three nucleotides downstream of the cleavage position strongly influence cleavage efficiency.** A) The putative hairpin in the *Sa-gapR* transcript, with the randomly mutagenised trinucleotide marked with red N's. The native RNase Y cleavage site is indicated by a light blue Y. B) The relative cleavage efficiency, measured with EMOTE, of each of the 64 possible combinations for the three nucleotides downstream of the RNase Y cleavage site. The wild-type trinucleotide (AGA) is shown in green and is set to 1 on the arbitrary scale. All other sequences were compared to the wild-type cleavage efficiency, using T-tests with correction for multiple testing. * = p<0.05, ** = p<0.01 and *** = p<0.001. The experiment was repeated four times. Trinucleotides mentioned in the text are highlighted with grey arrows. C) Northern blot of constructs with selected trinucleotides in sector III (pSaGap[IIIGGA], pSaGap[IIIAUU], pSaGap[IIICUA] and pSaGap[IIICGA]) showing the lack of cleavage with AUU. D) The potential for prolonging the hairpin stem when the trinucleotide is either AUU or GUU. R indicates a purine.

transcripts was examined by Northern blotting. No cleavage product was detected for the pSa-Gap[IIIAUU] transcript, whereas the other three transcripts were cleaved (Fig 9C).

Overall, most trinucleotide combinations gave transcripts that were cleaved more efficiently than the wild-type transcript, with some combinations (AUA, CUA, CUC, CUG and UUG) reaching >9-fold higher efficiency. The AUU and GUU trinucleotides can hypothetically extend the stem of the putative hairpin in sector IV, which would localise the cleavage site to the base of the stem and potentially prevent cleavage (Fig 9D). Both AUU and GUU transcripts are indeed cleaved with poor efficiency, and although AUU was not statistically different from wild-type in our EMOTE data, we confirmed the lack of cleavage by Northern blot (Fig 9C). Neither the CUU nor the UUU transcripts are cleaved with lower efficiency than the AGA transcript (Fig 9B), suggesting that a 2 base-pair extension of the stem is insufficient to block cleavage.

The base of the stem is one of the possible reference-points for the molecular ruler mechanism that help determine the cleavage position (Fig 6) and extending the stem by modifying Sector III will move the base of the stem further upstream, thus potentially shifting the cleavage

position (Figs 9D and S8A). However, when we examined the EMOTE data upstream of the native RNase Y cleavage, we observed no shift in cleavage position for trinucleotides that extend the stem by one or two base-pairs (NVU and YUU trinucleotides, respectively) (S8B and S8C Fig).

## Discussion

### Advantages of *in vivo* cleavage assays

RNase Y operates within the complex environment of the intracellular space and with the physical constraints inherent in its membrane localisation. Furthermore, the cell contains a large diversity of potential RNA substrates that can compete for binding to RNase Y. Of these, only an estimated ~15% are actually cleaved by RNase Y and, importantly, are not cleaved at random phosphodiester bonds, but rather at specific positions [5]. *In vitro* assays can generate precise kinetic measurements of the enzymatic cleavage, but such data might not be relevant in context of complex intracellular conditions. In this study we isolated total RNA from *S. aureus* cultures expressing different versions of the same transcript and then used Northern blotting to detect both the full-length and cleaved substrate. This provided preliminary quantification of *in vivo* cleavage efficiency for each transcript version but did not map the exact cleavage position. We therefore complemented the Northern blots with EMOTE assays, which not only identify the exact cleavage position for individual RNA molecules but also allow us to detect RNAs with low concentrations *in vivo* (Figs 1, 3, 4, 5, 6 and 8). However, it should be noted that due to lack of external normalisation, the targeted EMOTE protocol does not allow comparison of RNA abundance between two different cultures, leading us to present the EMOTE data as proportion of RNAs with a given 5'-end within a given window. The important exception to this limitation is when examining a single culture expressing a library of similar transcripts (Fig 9). Here EMOTE was used to identify each RNA molecule as well as the exact *in vivo* cleavage position, and since a normalisation was carried out on the DNA level of the library, the EMOTE data could be used to quantitatively compare the efficiencies with which each of the RNA types are cleaved.

We showed *in vivo* that sectors II-V of the *Sa-gapR* transcript are sufficient to promote an RNase Y cleavage in the otherwise non-targeted *fliM* mRNA (Fig 5). However, an unintended complication of using an *in vivo* assay consisted of a novel RNase Y-independent cleavage appearing in close vicinity to the native RNase Y cleavage position (Figs 5 and S9A). Nevertheless, the RNase Y-dependent cleavage of p*fliM*::II-V RNA means that sectors II-V contain all the elements needed to define a precise RNase Y cleavage position, but also that the exogenous p*fliM*::II-V RNA is presumably able to find RNase Y at the membrane. It is currently unknown whether this is caused by random diffusion of the mRNA or is facilitated by specific proteins.

The *in vivo* RNase Y cleavage site generated by sectors II-V (Fig 5), combined with the possibility of fine-tuning *in vivo* cleavage efficiency by single nucleotide changes in sector III (Fig 9), should make it possible to generate modules that can optimise gene expression profiles for synthetic biology projects in *B. subtilis*, *S. aureus* and possibly other bacteria that carry RNase Y. Further optimisation would be needed to eliminate the serendipitous cleavage site, but once accomplished a precise cleavage of any transcript can be designed to prevent translation, generate a small regulatory RNA or, as is the case in both the *Sa-gapR* and *Sa-glnR* operons, differentially express proteins encoded on the transcript.

### *In vivo* interchangeability of SaY and BsY

*B. subtilis* RNase Y (BsY) is fully functional in *S. aureus*, cleaving the pSaGap transcript at the same position as SaY, showing that the two RNase Y enzymes are sufficiently similar to exhibit

the same preferences for target RNAs as well as for the cleavage position on these RNAs (Fig 1D). Likewise, SaY is functional in *B. subtilis*, and cleaves *B. subtilis* RNAs in a similar manner to BsY (Fig 2).

RNase Y cleavage of the *Sa-gapR* operon and the *Bs-cggR* operon has been shown by DeLoughery and co-workers [19] to be dependent on RicF. In *B. subtilis* (and presumably also in *S. aureus*), RicT is loaded onto RNase Y by RicA and RicF [18]. Taken together, this strongly suggests that whatever requirement BsY has for RicA, RicF, and RicT in *B. subtilis* can be fulfilled by *S. aureus* RicA, RicF, and RicT, and vice versa, since SaY is functional in *B. subtilis* (Fig 2). Yet, this does not explain why RNase Y requires RicF for cleaving the majority of RNAs, while a minority of RNAs can be cleaved by RNase Y in absence of RicF [19]. However, since neither imprecise nor novel cleavage positions appeared in *B. subtilis* and *S. aureus ricF* mutants [19], they may not be involved in specifying the exact RNase Y cleavage position on the RNA molecules. Further studies will be needed to determine what transcript elements render RicF unnecessary.

## Cleavage efficiency factors

**Upstream factors.** Deleting the 12 nucleotides of sector II (leaving the G immediately upstream of the cleavage site) prevented cleavage (ΔII+G in Fig 4). We considered the possibility that this was because the sector II deletion brings the putative secondary RNA structure of sector I very close to the cleavage site, which potentially could prevent RNase Y access. However, when we removed the secondary structure by deleting both sector I and sector II, we did not restore cleavage, suggesting that sector II is required for cleavage. Furthermore, the sequence of sector II appears to be of little importance, since four very different mutant variants were all cleaved, and it is possible that a weak secondary structure upstream of sector I is what prevents cleavage in the pSaGap[ΔIΔII] and pSaGap[ΔIΔII+G] mutants (S5 Fig). Lastly, the importance (if any) of sector II seems to be confined to the SaGap transcript, since deletions of equivalent regions in other transcripts did not prevent cleavage (S5E and S5F Fig).

**Mutations in the sequence immediately downstream of the cleavage site can fine-tune efficiency.** Endoribonucleolytic cleavages can be maturation events, where the only objective of the cleavage is to obtain a final cleaved and stable RNA, or homeostasis-maintaining events, where the cleavage event switches the RNA from one function to another (or initiates the degradation of one or both of the generated fragments). Maturation events are expected to be highly efficient to avoid wasting energy on non-functional immature RNAs, whereas homeostasis-maintaining cleavage events are expected to be less efficient, since both uncleaved and cleaved RNA have a function for the cell. In our various Northern blots from wild-type cells, we are always able to observe both the uncleaved (full-length) and cleaved transcripts (Figs 1C and 2), suggesting that these cleavages are maintaining homeostasis. The ratio between full-length and cleavage product for the *Sa-gapR* transcript appears to have evolved to be in favour of the full-length (Fig 1C), and the native trinucleotide AGA found immediately downstream of the pSaGap cleavage site is indeed among the sequences that give the lowest cleavage efficiency (Fig 9B).

Transcripts with sector III trinucleotides that start with G (i.e. GNN) all appear to be cleaved with poor efficiency (Fig 9B). We do not believe that this is due to bias in the EMOTE protocol, since we have never observed an anti-G bias in the many other EMOTE assays carried out in our lab over the years. It is possible that the GNN trinucleotides shift the cleavage position one nucleotide downstream, which will prevent the EMOTE assay from attributing the cleaved molecule to a specific trinucleotide sequence. However, a Northern blot analysis of pSaGap[IIIGGA] revealed a cleavage product with low intensity, supporting the EMOTE data

(Fig 9C), and we therefore conclude that a guanosine immediately after the cleavage position lowers the efficiency of cleavage by RNase Y.

The only trinucleotides that result in lower cleavage efficiency than AGA are CGA, GGU, AUU and GUU. In contrast, there are many trinucleotides that increase cleavage efficiency, with some combinations (AUA, CUA, CUC, CUG and UUG) reaching >9-fold higher than the wild-type sequence. A common factor for these efficiently cleaved transcripts is that they contain a trinucleotide with U at the second position (NUN). However, this pattern is then modified by other factors, such as G at the first position (GUN trinucleotides are not efficiently cleaved) and whether the trinucleotide can extend the putative hairpin (e.g. AUU and GUU are the most inefficiently cleaved transcripts, presumably because they can both extend the putative hairpin).

**Secondary structures downstream of cleavage position.** The four model RNase Y targets examined in this study all have predicted hairpin structures downstream and in close vicinity to the cleavage position (Figs 4 and S9). We show that the least energetically favoured of the four hairpins, from the Sa-*gap* transcript, has duplexed nucleotides at the predicted positions *in vitro* (Fig 6), strongly suggesting that this hairpin forms *in vivo* as well. Furthermore, our mutational analyses of the putative hairpins in the pSaGap, pBsCgg and pSaGln transcripts revealed that the putative G-C base-pairs are crucial for cleavage efficiency, but that the orientation of the G-C base-pairs appear to be of little importance (Fig 7). However, inverting the remaining base-pairs of the hairpin stem almost completely abolishes cleavage (Fig 8). Taken together, this indicates that RNase Y cleavage of the examined transcripts requires a distinct primary nucleotide sequence downstream of the cleavage position, and that this distinct sequence must furthermore be in the context of a double-stranded RNA. However, our reporter system is *in vivo*, where the stem duplex might continuously form and then dissociate, and a given RNA molecule might therefore be cleaved or not depending on the current state of the duplex, and the RNA may indeed have evolved to have the optimal duplex/single-strand equilibrium to obtain the correct cleavage efficiency.

In contrast, the predicted loop-sequence of the hairpin does not appear to be of importance, since inversion of the loop does not detectably affect cleavage efficiency (S6B and S6C Fig). Elongation of the stem by two base-pairs has likewise no detectable effect on cleavage efficiency, but if the stem is extended by three base-pairs, then cleavage efficiency is reduced drastically (Fig 9B, 9C and 9D) (Note that these effects could also be directly due to the changes in the primary sequence needed to obtain such stem extensions).

Remarkably, although our results agree with previous studies on the importance of putative duplex formation downstream of the cleavage site [14,21], there are multiple examples from *S. aureus*, *S. pyogenes* and *B. subtilis* of RNase Y cleavage sites that lack predicted downstream duplex formation [5,19,20,24], and we suspect the existence of two or more RNase Y targeting pathways.

## Redundancy between sectors V and VI

Deleting either sector V or sector VI individually had no detectable impact on cleavage efficiency. However, when both sectors were deleted, then it became impossible to detect any cleavage products by Northern blotting, and it required the highly sensitive EMOTE assay to detect any cleavage at all. It thus appears that sector V and VI both contain sequence elements that promote RNase Y cleavage, but that these elements are mutually redundant.

## Cleavage positioning factors

Seen in the light of global RNA decay, the precise positioning of an endoribonucleolytic cleavage can seem unimportant, since almost any cleavage will accomplish the goal of providing

access-points for exoribonucleases to finish the degradation. Such low fidelity was indeed observed for *E. coli* RNase E, where the cleavage motif is gnwUu and each RNA molecule is cleaved multiple times [27]. RNase Y appears to have more stringent cleavage criteria, since transcriptome studies in *S. aureus*, *B. subtilis* and *S. pyogenes* identified a limited number of RNase Y cleavage positions (99, 27 and 190 positions, respectively) [5,19,20]. Nevertheless, a recent study showed that some of the phenotypes of a *B. subtilis* RNase Y mutant can be complemented by expressing RNase E from *E. coli*, and that this complementation was highly dependent on anchoring RNase E to the membrane [28]. Importantly, these complemented strains were unable to properly degrade the *Bs-cggR* transcript and no version of RNase E could generate the stable downstream cleavage product observed in this and other studies [21,22,28]. This suggests that RNase E can only complement the general RNA decay function of RNase Y, but for normal (rapid) growth, *B. subtilis* appears to require the type of controlled efficiency and precise positioning of cleavage events that we examine in this study.

## The upstream nucleotide fine-tunes the site selectivity

A guanosine has been observed upstream of most RNase Y cleavage sites in *S. aureus* and in *S. pyogenes* this appears to be a requirement for cleavage [20,24] and more recently in *B. subtilis* [29]. However, our results show that for *S. aureus* and *B. subtilis* RNase Y (at least for the *Sa-gapR* transcript) this is only a preference and not a strict requirement, and that the main role of the guanosine is to direct all RNase Y cleavage events towards the same phosphodiester bond (Fig 3). Such precision is presumably of little importance if the cleavage products are destined for immediate degradation, but precision might be advantageous in situations where one or both of the two resulting RNA cleavage products should have ends that protects them against exoribonucleases.

## A reference-point for a molecular ruler mechanism is located within sector IV

Deletion of the three nucleotides in sector III caused the cleavage site to shift upstream by four nucleotides (Fig 4), indicating that the RNase Y cleavage site is selected by measuring a defined distance from a specific reference point downstream of the cleavage site. The 1 nt difference could be due to an imprecise ruler mechanism and/or to the presence of a non-preferred uridine at three nucleotides upstream of the native site (position +255). Since deletion of sector V does not shift the cleavage position (Fig 4C) we conclude that the reference point for the molecular ruler must be within sector IV (Fig 10). This is consistent with the previously proposed molecular ruler mechanism where RNase Y consistently cleaved the *saePQRS* transcript 6 nt upstream of a putative hairpin stem (Rrs-duplex), but it was not examined what aspects of the Rrs-duplex determined the cleavage position [14].

Our data from sector III mutants that are predicted to move the stem-bottom one or two nucleotides upstream (NVU and YUU trinucleotides, respectively) all exhibit clear major cleavage positions at the native position (S8 Fig), and we therefore conclude that the transition from single stranded to duplexed RNA is not the reference point for the molecular ruler mechanism.

This conclusion is supported by the exchange of hairpins between SaGap and BsCgg (pSaGap[cggHP]), where the distance from native *Sa-gap* cleavage position and the beginning of the stem was maintained at three nucleotides but the cleavage position was shifted two nucleotides downstream (green line in Fig 6F), strongly suggesting that elements within the *Sa-gapR* and *Bs-cggR* hairpins serve as reference point for the ruler mechanism.

One of these elements is potentially U280 on the downstream "hairpin-arm", which when mutated to an A (pSaGap[U280A]) causes RNase Y to cleave two nucleotides further

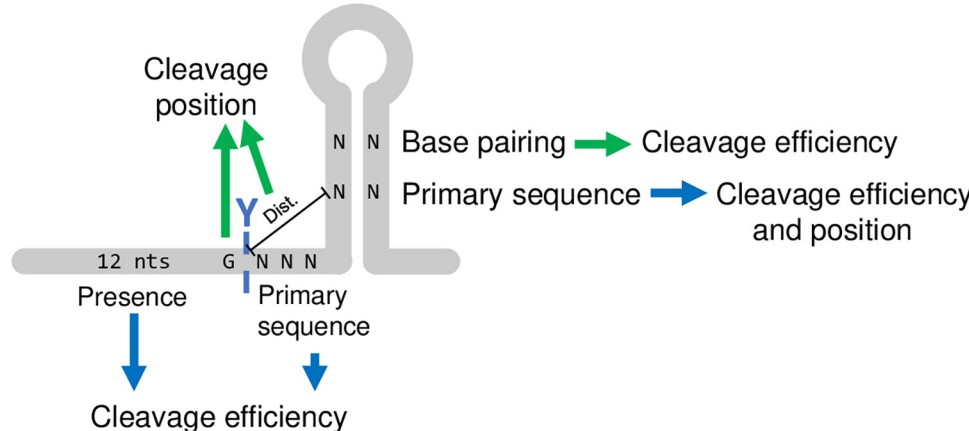

**Fig 10. Schematic summary of elements defining an RNase Y cleavage site.** Blue arrows denote effect that were identified in the SaGap transcript and green arrows denote effect that were also identified in other transcripts. A hairpin structure is often found a few nucleotides after the RNase Y cleavage position. Cleavage efficiency is defined by the primary sequence of the three nucleotides after the cleavage position as well duplex-formation in the downstream hairpin. The region upstream of the cleavage (sector II) is also important for efficiency, but its sequence appears to be unimportant. Cleavage positioning is defined by a G right before the cleavage position as well as distance to sequence elements within the stem of the downstream hairpin.

downstream (Fig 8). This effect appears to be unrelated to duplex-formation, since mutating the base-pairing A264 on the upstream "hairpin-arm" to a U (pSaGap[A264U]) does not shift the cleavage position (Fig 8).

## A model for RNase Y targeting

Taken together, our data shows that the region surrounding the RNase Y cleavage site in SaGap has all the factors required for both targeting by RNase Y and cleavage at a precise position. Moreover, different elements control the cleavage efficiency and the cleavage positioning. The downstream hairpin plays a key role, since base-pairing in one part of its stem is essential for efficient cleavage (Fig 7), whereas other specific sequence elements (but not their base-pairing) are important for positioning the cleavage (Fig 8). Only the length of the sequence between the cleavage position and the hairpin is important for positioning (Fig 4), but its composition can modify the cleavage efficiency more than 20-fold (Fig 9). While the SaGap transcript was the main focus, we have demonstrated several of the above-mentioned elements in other transcripts from both *S. aureus* and *B. subtilis* (summarised in Fig 10).

## Materials and methods

### Growth of strains and media

Unless stated otherwise, *S. aureus* was grown in MHB liquid medium (Mueller Hinton Broth, Becton Dickinson) or agar plates at 37°C. MH medium was systematically supplemented with uracil (20 mg/l) as the parental strain PR01 is Δ*pyrFE* [30], and 10 mg/l chloramphenicol was used to select for transformants and maintain vectors. *B. subtilis* was grown in 2XTY at 37°C.
*Escherichia coli* was grown in LB medium with 100 mg/l ampicillin to select for plasmids.

### Vector design and construction

The *Sa-gap/Bs-cgg* and *Sa-gln/Bs-gln* operon pairs were identified as evolutionary conserved RNase Y targets by aligning (using protein BLAST) the proteins encoded downstream of the

RNase Y cleavage sites described by DeLoughery and co-workers [19] with the translated *S. aureus* genome. The genome positions of the *S. aureus* hits were then compared to the list of *S. aureus* RNase Y cleavage sites described by Khemici and co-workers [5].

All *S. aureus* expression vectors constructed in this study are based on plasmid pEB01, an *E. coli-S. aureus* shuttle-vector which carries a transcription terminator sequence downstream of the polylinker [31]. pSaGap and pBsCgg were constructed by PCR-amplifying the regions indicated in Fig 1 with primers that added restriction sites for SalI, KpnI and AscI (See S2 Table). After digestion with the relevant restriction enzymes, the inserts were ligated into the SalI and AscI sites of pEB01 and transformed into *E. coli*. pSaGln and pBsGln were constructed using In-Fusion cloning assembly (Takara). Specific modifications in the vector sequences were introduced via divergent PCR with 5' phosphorylated primers, followed by blunt-end ligation with T4 DNA ligase (Thermo Fischer) and transformation into *E. coli*. All vectors-inserts were verified by Sanger sequencing before transformation into *S. aureus* strains WT and ΔY (and BsY where relevant).

Pl799: pHM2-rny (BsY gene under control of its own promoter cloned in pHM2 for integration in *amyE* locus)

BsY was amplified from *B. subtilis* DNA with oligo pair CC2150 / CC2151 and cloned in pHM2 plasmid (EcoRI/SalI) for integration at the *amyE* locus.

Pl800: pHM2-cvfA (SaY gene cloned under promoter and SD control of BsY. Replacement of BsY ORF by SaY ORF.

SaY was amplified from *S. aureus* DNA with oligo pair CC2228/CC2155. The 5' and the 3' UTR of BsY from *B. subtilis* DNA with oligo pairs CC2150/CC2229, and CC2154/CC2151. The three overlapping PCR fragments were assembled in new PCR reaction with oligo pair CC2150/CC2151.The fragment was cloned in pHM2 plasmid (EcoRI/SalI) for integration at the *amyE* locus.

Please see S1 Table for strain-list.

## Constructing random library of Sector III

The three base-pairs of pSaGap Sector III were randomised by polymerase chain reaction (PCR) using the pSaGap construct as template and primers 634 and 636 listed in S2 Table. The resulting plasmid library of 64 ($4^3$) potential variants was transformed first into *E. coli* DH5α then into *S. aureus* WT (generating WT+pSaGap[IIIrandom]). More than 6400 CFUs were recovered from each transformation to ensure that all 64 variants are represented in the library.

## RNA extraction

*S. aureus* cultures were inoculated from overnight cultures prepared from a single colony and diluted at 1/100 into fresh medium, 8 ml culture were harvested at $OD_{600} \sim 0.4$ and immediately mixed with 40 mL ice cold ethanol–acetone mix (1:1). After centrifugation (10 min at 3500g), the cells were washed once in 1ml TE buffer (10 mM Tris pH8, 1mM EDTA) and suspended in a mix of 90 μL TE buffer + 10 μL lysostaphin 35520 U/ml (EUROMEDEX) + 2,5 μL RNasin Plus 40 U/μl (Promega). Samples were incubated at 37°C for 10 min before addition of 300 μl TRI-Reagent (MRC). A Phase Lock Gel tube (5Prime) was used to obtain a pure aqueous phase, in which 20 μg of glycogen (Roche) and 150 μl isopropanol was added to precipitate the RNA over-night at -20°C. RNA pellets were washed with 900μL cold 75% ethanol, re-suspended in 20μL 0.1X TE and stored at -80°C.

*B. subtilis* cultures were inoculated from overnight cultures prepared from a single colony and diluted at 1/500 into fresh medium, 1,4 ml culture were harvested at $OD_{600} \sim 0.6$ (time

point 0 min) and 2.5, 5, 10, 15, 20, 25 and 30 min after rifampicin was added to block transcription. The harvested cultures were mixed with 100µl of ice cold 10 mM sodium azide (NaN₃) and centrifuged at 4°C (1 min at 13000 rpm). The pellet was immediately frozen in dry ice. RNA was isolated by the RNAsnap method described in [32].

### Analyses of RNase Y cleavage events

The cleavage (or lack thereof) was studied on total RNA isolated from *S. aureus* strains carrying vectors expressing the various transcripts of interest (S1 Table).

### Northern blotting

For *S. aureus* RNA the Northern blotting was carried out as described in [5]: Briefly, 4µg total RNA were separated by PAGE in a 6% polyacrylamide 8M urea gel, transferred to a Hybond-N+ membrane (Amersham), then UV crosslinked 120 J/cm$^2$ (Hoefer, UVC 500). The size marker (RiboRuler Low range, Thermo Scientific) as well as ribosomal RNAs were visualised with Methylene blue. Blots were hybridized with $^{32}$P labelled synthetic DNA probes P1, P2 or 5S (S2 Table) and revealed using a phosphorimager (Fuji).

For *B. subtilis* RNA: 5µg total RNA were separated on 1% agarose gel transferred by capillarity to a Hybond-N+ membrane (Amersham) in 5X SSC, 0.01M NaOH, then UV crosslinked 120 J/cm$^2$ (UVP, HL2000). Hybridization was performed using 5'-labeled oligonucleotides *atpB* (CC2437) and *glnA* (CC2438) using Ultra-Hyb (Ambion) hybridization buffer at 42°C for a minimum of 4 hours. Membranes were washed twice in 2XSSC 0.1% SDS (once rapidly at room temperature (RT) and once for 10 min at 42°C) and then 3 times for 10 min in 0.2XSSC 0.1% SDS at RT. The CC058 probe against 16S rRNA was used as loading control. Quantification was done with Image J software.

### Targeted EMOTE

Targeted EMOTE is a variation of the original EMOTE described in detail previously [23]. Specificities of the targeted EMOTE are described here.

Ligation was carried out using 4 ug of total RNA with 1 ul of 100uM BioRp8 in a 10 ul volume and placed at 85°C for 2 minutes and flash cooled in ice water. A mix of 3µl water, 2 µl of 10X RNA ligase buffer (NEB), 2 µl 10 mM ATP, 1 µl of RNasinPlus 40 U/µl (Promega), 1µl RNA ligase 10 000 U/µl (NEB) and 1 µl RppH 5000 U/µl (NEB) was added to the RNA/oligo mix. Ligation was performed in the presence of RppH in order to detect the primary transcripts for normalisation purposes. The ligation mix was placed for 3 hours at 37°C. RNA from the sample was purified by ethanol precipitation.

Reverse transcription was carried out as described in [23] using the plasmid-specific pEB-TT-R3-illuB primer to specifically reverse transcribe the exogenous transcripts.

### Illumina sequencing and analyses of EMOTE data

Illumina sequencing consisted of 50 nucleotide single-end sequencing, except for the results presented in Fig 9 where 100 nt reads were required. The sequencing was carried out at the genomics platform of University of Geneva (https://lims.ige3.genomics.unige.ch/). An R script was used to process the raw sequencing data: Each read was demultiplexed, using EMOTE barcodes to assign each read to its original sample. Reads were then mapped with Bowtie onto the sequence of the vector that had provided the examined transcript (for example pSaGap, pBsCgg, pSaGap[InvStem], etc.). A table was then generated which indicates the exact 5' nucleotide position of each detected RNA molecule [33].

In order to avoid bias generated by preferential PCR amplification of one cDNA over another, the EMOTE protocol employs a Unique Molecular Identifier (UMI) in the oligo that is ligated to the 5' ends of the RNA samples. The UMI has 2187 ($3^7$) possible combinations, and since all experiments generated > 5000 reads for a given RNA cleavage (and usually much more than 5000 reads), we used a subsampling (also referred to as down-sampling) approach to get below the saturation level of the UMI. Briefly, for each position analysed in this work, if the number of UMIs was >150, data was subsampled by a 'dilution factor' until the output was in the 50–150 interval. Subsampled UMI counts were then multiplied by the "dilution factor" in order to get a calculated UMI count for the original sample. This UMI count then represents the number of detected individual RNA molecules with a given 5' end.

For each EMOTE graphical representation presented, we calculated the total number of detected RNA molecules with a 5' end within the shown bracket. Each column indicates the percentage of individual RNA molecules with a 5' end at the given position. A tall column for nucleotide 'X' therefore indicates that 'X' was the 5' nucleotide in a majority of RNA molecules.

### Target-specific DMS-MaPseq of in vitro folded *Sa-gap* mRNA

Template for in vitro transcription was amplified by PCR using primers 835 and 463. A 530 nt *gap* transcript was produced using reagents from Thermo Scientific following the protocol for conventional *in vitro* transcription, three RNA samples were independently generated and treated in parallel in the following steps. RNA was purified by phenol-chloroform after 30 minutes DNase I treatment. Transcript size was verified on 8% denaturing polyacrylamide gel and RNA was resuspended in 1x TE buffer. 350 ng of *gap* transcript was denatured 3 minutes at 90˚C in 25 uL of DMS buffer (10 mM Na-cacodylate pH 7.4, 0.1 mM EDTA, 100 mM KCl) and cooled down at room temperature for 15 minutes before 30 minutes folding at 37˚C in the presence of 10 mM MgCl2. 1 uL of a 1:12 DMS dilution in ethanol was added and incubated for 3 minutes. DMS was quenched by adding 475 uL of quenching solution (4.29 M β-mercapto-ethanol, 0.3 M sodium acetate) and the modified gap transcript was concentrated by ethanol precipitation. First strand cDNA was synthetized with TGIRTIII enzyme (Ingex) and 1 μM primer 868 for 2 hours at 57˚C. After precipitation, cDNA libraries were amplified by 31 cycles of PCR with primers to introduce Illumina adapters and sample barcodes (primers 233, 255, 259, 260 and 815–818, S2 Table). The resulting PCR products were purified and pooled to be sent for Illumina sequencing with 100 nt read-length.

After demultiplexing, reads quality was checked using FastQC before being mapped to the reference *gap* mRNA sequence using Bowtie2 with default parameters. Rsamtools was used to count nucleotides frequency at each position. Data file manipulations and graphics were generated with R, scripts are available on request.

### Analysis of trinucleotide variants in Sector III

Total DNA was isolated from *S. aureus* WT cultures carrying the pSaGap[IIIrandom], and PCR was used to amplify the trinucleotide locus from the library cultures (in two biological replicates). The PCR products were Illumina sequenced and the relative abundances of each of the 64 expression vectors in our libraries were determined by counting the number of reads with each trinucleotide combination (AAA, AAC, etc.).

Total RNA was also isolated from *S. aureus* WT cultures carrying the pSaGap[IIIrandom], and used for targeted EMOTE sequencing (in four biological replicates). The EMOTE sequencing reads were sorted into 64 separate files according to the trinucleotide they contain (AAA, AAC, etc.). For each set of reads, the calculated number of individual RNA molecules

(see above) at the native RNase Y cleavage position (+259) was normalised with the relative abundance of each corresponding expressing vector in the library (e.g. number of RNA molecules with AAA / number of vectors with AAA).

A Bilateral paired Student test (T-Test) was then performed to determine whether RNase Y cleaved the transcript with a given trinucleotide with a different efficiency than the transcript with a WT trinucleotide sequence (AGA). The calculated p-values were then adjusted with Bonferroni correction for multiple testing.

## Supporting information

**S1 Table. List of strains.**
(DOCX)

**S2 Table. List of oligos.**
(DOCX)

**S1 Fig. Alignment of *S. aureus* and *B. subtilis* RNase Y. SaY: *S. aureus* RNase Y, BsY, *B. subtilis* RNase Y, HD: the two critical amino acids in the active site.** Adapted from Redder, 2018.
(DOCX)

**S2 Fig. PNPase degrades the upstream cleavage fragment of the pSaGap transcript.** A) The layout of the pSaGap construct, with the location of the upstream and downstream Northern blot probes (P2 and P1, respectively). B) Northern blot showing the appearance of two new bands when using the P2 (upstream) probe on total RNA from a PNPase deletion strain carrying pSaGap. Black and grey triangles indicate full-length transcript and cleavage products, respectively. This data is in agreement with a large-scale *Streptococcus pyogenes* study where upstream products of RNase Y cleavage are systematically degraded by PNPase [20].
(DOCX)

**S3 Fig. EMOTE data from the ΔY strain corresponding to Figs 1, 4, 6 and 8.** The EMOTE data is presented as proportions of RNA molecules with a given 5' end on the Y-axis (number of reads detected at a specific position divided by the total number of reads detected within the chosen window). Note that the number of detected molecules with 5'ends within the shown windows is much larger in the WT strain than in the ΔY strain, since RNase Y does not cleave the RNAs in the ΔY strain. The ΔY data is therefore based on a very low number of detected molecules and corresponds to background noise. The random nature of this noise can sometimes lead to a tall column for a position, since the EMOTE data is presented as proportions, with sum of the columns set to 1 (this is for example the case for the 6th position in the ΔY pSaGln data on panel A, where no cleavage is observed in the Northern blot in Fig 1C). The native RNase Y cleavage position is indicated with blue dotted lines. Green dotted lines indicate shifted cleavage positions where relevant. A) Corresponds to Fig 1. B) Corresponds to Fig 4C. C) Corresponds to Fig 4D. D) Corresponds to Fig 6. E) Corresponds to Fig 8.
(DOCX)

**S4 Fig. Translation does not influence RNase Y cleavage.** A) Northern blot showing RNase Y dependent cleavage in pSaGap with a frame-shift (pSaGap[Δ128]) and with the start-codon mutated (pSaGap[NoStart]). B) EMOTE data confirming that the frame-shift in pSaGap [Δ128] has not shifted the RNase Y cleavage position.
(DOCX)

**S5 Fig. Mutations upstream of the RNase Y cleavage sites.** A) Northern blot for the deletions of sectors I and II of pSaGap, with or without the G upstream of the cleavage position. B) Putative secondary structure that can form if both sectors I and II of pSaGap are deleted. Native RNase Y cleavage positions are indicated with blue dotted lines. Sector III is highlighted in green. C) The sequences of the wild-type pSaGap sector II and the four mutated versions. The G immediately upstream of the cleavage site has been underlined, and the nucleotides that differ from the wild-type sequence are shown in red. D) Northern blot showing that while the transcript from pSaGap[ΔII+G] is not cleaved, the transcripts from all four mutant variants are cleaved. The strain background is shown below each lane. The full-length and cleaved transcripts are indicated by the black and grey arrowheads, respectively. E) Northern blot showing cleavage of pBsCgg and BsCgg[Δ12] (where 12 nucleotides immediate upstream of the conserved G have been deleted). F) Northern blot showing cleavage of pBsGln and BsGln[Δ12] (where 12 nucleotides immediate upstream of the conserved G have been deleted). A weak band corresponding to the cleaved product has been marked with an asterisk.
(DOCX)

**S6 Fig. Putative hairpin in pBsGln and inversion of the putative hairpin loop in Sector IV of pSaGap.** A) The putative hairpin immediately downstream of the RNase Y cleavage in pBsGln. B) The sequence of the putative InvLoop hairpin where the loop has been modified so that U becomes A and A becomes U (mutated nucleotides shown in red). C) Northern blot showing the cleavage of the pSaGap[InvLoop] transcript.
(DOCX)

**S7 Fig. EMOTE data for hairpins with mutated G-C base-pairs, corresponding to Fig 7A and 7C in the main text.** Note that the EMOTE data is presented as proportions of RNA molecules with a given 5' end on the Y-axis (number of reads detected at a specific position divided by the total number of reads detected within the chosen window), and that the number of detected molecules is much larger in the WT strain than in the ΔY strain.
(DOCX)

**S8 Fig. Extending the hairpin stem does not alter the cleavage position.** A) Wild-type (AGA) *Sa-gapR* hairpin, hairpin with a single base-pair extension (NVU) and hairpin with two base-pairs extension (YUU). Y: Pyrimidine bases, V: A, C or G. Red nucleotides are varied and green U's are the uridines that extend the putative hairpin stem by one or two base-pairs (NVU and YUU, respectively). B) EMOTE data from NVU trinucleotide combinations that can potentially extend the stem by a single base-pair. The cleaved nucleotide sequence is shown below the graph and the native cleavage position is shown by a light blue dotted line. Red nucleotides are varied and the green U's extend the putative hairpin stem by one base-pair. C) EMOTE data from YUU trinucleotide combinations that can potentially extend the stem by two base-pairs. The cleaved nucleotide sequence is shown below the graph and the native cleavage position is shown by a light blue dotted line. Red nucleotides are varied and the green U's are the uridines that extend the putative hairpin stem by two base-pairs.
(DOCX)

**S9 Fig. mFold predictions of secondary structures surrounding the RNase Y cleavage sites.** RNase Y cleavage sites are indicated with blue arrows. Structure prediction and free energy calculations were performed on the mFold.org server using default settings (Zuker 2003). A). The pfliM::II-V transcript. The RNase Y independent cleavage position is indicated in purple. B). pBsCgg transcript. C). pSaGln transcript. pBsGln transcript.
(DOCX)

## Acknowledgments

We would like to thank Manuel Campos for scientific discussions, Pierre Genevaux and Raffaele Ieva for critical reading of the manuscript, and the iGE3 Genomics Platform at University of Geneva, Switzerland, for help and patience with development of novel sequencing protocols.

## Author Contributions

**Conceptualization:** Alexandre Le Scornet, Sylvain Durand, Ciarán Condon, Peter Redder.

**Formal analysis:** Alexandre Le Scornet, Ambre Jousselin, Kamila Baumas, Sylvain Durand, Peter Redder.

**Funding acquisition:** Peter Redder.

**Investigation:** Alexandre Le Scornet, Ambre Jousselin, Kamila Baumas, Gergana Kostova, Sylvain Durand, Leonora Poljak, Eve Coutant, Romain Pigearias, Gabriel Tejero, Jonas Lootvoet, Céline Péllisier, Gladys Munoz, Peter Redder.

**Methodology:** Ambre Jousselin, Peter Redder.

**Project administration:** Peter Redder.

**Software:** Ambre Jousselin, Roland Barriot.

**Supervision:** Sylvain Durand, Ciarán Condon, Peter Redder.

**Visualization:** Alexandre Le Scornet, Ambre Jousselin, Sylvain Durand, Peter Redder.

**Writing – original draft:** Alexandre Le Scornet, Ciarán Condon, Peter Redder.

**Writing – review & editing:** Alexandre Le Scornet, Ambre Jousselin, Kamila Baumas, Sylvain Durand, Leonora Poljak, Ciarán Condon, Peter Redder.

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
