## [Decision Letter · Decision Letter 0]

31 Mar 2024

Dear Dr Redder,

Thank you very much for submitting your Research Article entitled 'Critical factors for precise and efficient RNA cleavage by RNase Y in Staphylococcus aureus' to PLOS Genetics.

The manuscript was fully evaluated at the editorial level and by independent peer reviewers. The reviewers appreciated the attention to an important topic but identified some concerns that we ask you address in a revised manuscript.

We therefore ask you to modify the manuscript according to the review recommendations. Your revisions should address the specific points made by each reviewer.

Yours sincerely,

Kai Papenfort

Academic Editor

PLOS Genetics

Lotte Søgaard-Andersen

Section Editor

PLOS Genetics

Dear Dr. Redder.

Thank you again for submitting your work to PLOS Genetics. We have now received the comments from the reviewers of your manuscript. Overall, all three reviewers were very positive about your manuscript, however, they also requested several changes to the text, as well as few additional experiments. For example, reviewer #1 requests to replicate the results with at least one other operon to support the general relevance of the results.

Please make sure to respond to all the comments in your revised manuscript and the rebuttal letter.

Best wishes,

Kai Papenfort

(Academic Editor)

Reviewer's Responses to Questions

**Comments to the Authors:**

Reviewer #1: Rny is a membrane-localized RNase, which is thought to be responsible for the initiation of bulk mRNA turnover in firmicutes. However, Rny cleaves only a limited number of mRNAs. Cleavage may result in stabilisation of the processed mRNAs thereby playing an important role in regulation of co-transcribed genes. It is debated how Rny specificity is determined, and no clear processing site (besides a conserve G resiue) could be identified. Rny activity is likely determined by secondary structure recognition determinants, downstream of the cleavage site.

Here the authors analysed Rny dependent operon processing of two homologous transcripts present in B. subtilis and S. aureus. They could show that Rny dependent processing is highly similar in both species. Through thorough mutant analyses they could show that cleavage is dependent on secondary structure in close vicinity to the processing site. The results were gained through Northern blot analysis and startsite predictions using EMOTE. This is a well-written manuscript with well suited experiments.

Results are mainly based on the SaGap operon and northern blot/EMOTE analyses. It would be more convincing if the main results could be reproduced in at least one more operon.

Specific comments

Line 57: reference Meinken et is not appropriate for the statement

Line 77: Add reference for the statement on the membrane anchor

Line 90: The interaction of RNaseY with auxillary proteins (YlbF, YmcA, and

YaaT (the Y-complex) should be introduced here.

Line 225: add a reference to the Fig.

Figure 4: The meaning of the asterisk in Fig 4C is not clear. Why are the results not shown.

Line 374: The analysis of the SaGap operon is very extensive and conclusive. It would be very helpful to analyse the other operons for similar structures (e.g. the loop of sector IV)? The validation of at least one of these structures would be very helpful .

Line 683: It remains speculative why and how this upstream sequence (sectorII) would be important. Is a similar upstream region also important in the other target genes.

Line 725: Show the predicted structures of all 4 transcripts.

Reviewer #2: RNase-driven decay of RNA is a universal and essential process for all living organisms that rapidly adapts transcript availability to current needs and also mediates the processing and maturation of cellular RNAs. The present study focusses on the endoribonuclease RNase Y of the Gram-positive bacterium S. aureus and the factors determining its cleavage function.

The authors opted for an in vivo system in which a native S. aureus and a corresponding RNaseY deletion mutant were used as the experimental backgrounds for cleavage of model transcripts provided on a vector. In an impressive series of elegantly conceived and meticulously conducted experiments, the authors identified the sequence and structural constraints that determine RNase Y target site selection and cleavage efficiency. Also, they demonstrate that RNases Y from S. aureus and B. subtilis can functionally replace each other in vivo. This is a comprehensive study, exhaustive down to the smallest detail, which builds on much previous knowledge to provide an accurate picture of how RNase Y selects and cleaves its RNA targets.

The manuscript is very well written and leads the reader logically and step by step through the experiments and their theoretical considerations as well as the results. I would like to emphasise that it was a real pleasure to follow this scientific endeavour. However, this strength of the manuscript is also its weakness.

1. Although all data and details were obtained and can be found in the paper, they are not summarised at any point in a concise and comprehensible manner. The authors should definitely look for a way to rectify this, for example by revising the abstract and also by adding a section (either in the results section or the discussion) in which they clearly identify the sequence and structural features that determine RNaseY targets. Figure 10 in the Discussion would be a good starting point to build on. Otherwise, there is a risk that the data cannot be adequately utilised by the scientific community.

More specifc comments:

2. Figure 1D: Is this deceptive or does the pSaGap allow a little more leeway regarding the cleavage site than the other transcripts? It seems that there are a little bit more alternative cleavage sites (?)

3. Figure 4 B and C, last panel (pSaGap sector V/VI deletion). In the Northern blot, cleavage is obviously impaired in this mutant, but the EMOTE data (Fig 4 D) still show cleavage at the correct position. How can this be explained?

4. pg. 13, experiment on the exogenous fliM transcript: "We could detect the full-length fliM transcript in both S. aureus WT and delta Y strains". In Figure 5B left, I can only see a signal in the RNase Y mutant background, but not in the wildtype.

5. Further to the experiment on the exogenous fliM transcript: Did actually the surrounding fliM RNA of the gap II-V insertion influence the folding of the RNase Y target?

6. pg. 20 line 577 "The wild-type trinucleotide sequence (AGA) turned out to be one of the worst sequences for RNase Y cleavage efficiency". Did this sequence evolve/prevail to protect S. aureus RNAs from excessive RNA cleavage? If so, why are other poor-cleaving triplets under-represented as targets?

Reviewer #3: The ribonuclease RNase Y is a player in RNA metabolism in Firmicute bacteria, and in this study, the authors have explored the determinants for RNase Y activity on target RNAs. The authors show that the nucleotide sequence surrounding an RNase Y cleavage site is sufficient for cleavage and demonstrate that a non-substrate RNA can be converted into a substrate. They use mutational analyses to determine which nucleotides in the vicinity of the cleavage are required for efficient cleavage and which nucleotides determine the exact cleavage position. These findings are shown to operate in both S. aureus and B. subtilis and are likely to be general within the Firmicutes. They present evidence that the S. aureus gapR transcript has evolved to maintain a precise cleavage position, but with low cleavage efficiency, and that the efficiency can be modulated by sequence changes in the RNA. One striking result is that they mapped the RNase Y recognition within the GapR operon and showed that this region can be inserted into a different RNA molecule and convert it into an RNase Y target. The study is well conducted and the results clearly explained. The authors mainly utilized in vivo approaches, such as Northern blotting, and EMOTE, to elucidate the mechanism of RNase Y cleavage site and the secondary structural requirement of transcripts for the catalysis. The results and reasoning are mostly clear and understandable, but in vitro experiments (in the future) might be also helpful to support some of their conclusions. There are a few other minor comments that will hopefully be helpful for the authors to consider:

Interesting new reference with might be added at line 37? Jenniches et al (2024) Improved RNA stability estimation through Bayesian modelling reveals most Salmonella transcripts have subminute half-lifes. 121, e2308814121.

Line 144 ‘argued’ might be better replaced with ‘reasoned’?

Page 11, line 328-329

Investigation of mutation in conserved bases in the sector II between B. subtilis and S. aureus would be interesting. It might be interesting if the authors can try test cleavage pattern with a simplified system in vitro.

Page 11, line 337-339

The disappearance of cleaved band caused by the truncation of sector V and VI may also be merely due to the lack of scaffold for RNase Y to be recruited on the RNA. This can be tested by substituting these sectors with a long nucleotide which does not form any secondary structures.

Page 13, line 390-394

It is very interesting that an exogenous RNA (E. coli fliM) could be transformed into a target for RNase Y by adding regions II-V of SaGap. Is there a hypothesis on what other endoribonuclease might be getting recruited and cleaving in the vicinity and why this is not the case in SaGap? Does the negative control using the RNase Y knockout strain in this experiment eliminate the possibility that an unknown ribonuclease cleaved the transcript from pfliM::II-V at both canonical and 2nt upstream of canonical sites instead of RNase Y ? Could an unknown nuclease work dependently on RNase Y?

Page 15, line 394

The reference of the figure should be 3D.

Page 18, line 542

The numbering of mutated base (U280A) does not match with that displayed in the figure 8A (U282A). This is the case with other sentences in this manuscript.

Page 23, line 684-689

Could deletion of both sector I and II result in putting another hairpin structure (which may be formed the sequence upstream of these sectors) in proximity to the cleavage site and thereby occluding it? This may need to be tested and mentioned in this article.

Page 25, line 765-768

This explanation of why the precise cleavage can be advantageous might need rephrasing for clarity.

Figure 5: The levels for fliM in the WT background seem lower than in the ΔY strain.

Figure 8 and lines 541-544: Can the order of the A264U and U282A mutants in Figure 8 be swapped to match the order they are mentioned in the text?

Figure 9: Could they highlight those trinucleotide combinations that are mentioned in the text?

**Have all data underlying the figures and results presented in the manuscript been provided?**

Reviewer #1: Yes

<

---

## [Decision Letter · Decision Letter 1]

23 Jun 2024

Dear Dr Redder,

We are pleased to inform you that your manuscript entitled "Critical factors for precise and efficient RNA cleavage by RNase Y in Staphylococcus aureus" has been editorially accepted for publication in PLOS Genetics. Congratulations!

Yours sincerely,

Kai Papenfort

Academic Editor

PLOS Genetics

Lotte Søgaard-Andersen

Section Editor

PLOS Genetics

Comments from the reviewers (if applicable):

Dear Dr Redder.

I have now received feedback from the referees on your revised manuscript and I am happy to inform you that all three suggested to accept it (and so have I).

Congratulations on a very nice manuscript.

Best wishes,

Kai Papenfort

Reviewer's Responses to Questions

**Comments to the Authors:**

Reviewer #1: With the new data the manuscript is now very convincing. I have no further suggestions

Reviewer #2: The authors have done a tremendous job to further improve this very interesting manuscript. They have sufficiently addressed my previously raised issues and I have no further comments regarding this submission.

Reviewer #3: The authors have provided compelling responses to the reviewer comments and the updated manuscript has been improved.

**Have all data underlying the figures and results presented in the manuscript been provided?**

Reviewer #1: Yes

Reviewer #2: Yes

Reviewer #3: Yes

PLOS authors have the option to publish the peer review history of their article (what does this mean?). If published, this will include your full peer review and any attached files.

Reviewer #1: **Yes: **Christiane Wolz

Reviewer #2: No

Reviewer #3: No

**Data Deposition**

http://datadryad.org/submit?journalID=pgenetics&manu=PGENETICS-D-24-00279R1

**Press Queries**

---

## [Editor Report · Acceptance letter]

26 Jul 2024

PGENETICS-D-24-00279R1 

Critical factors for precise and efficient RNA cleavage by RNase Y in *Staphylococcus aureus*

Dear Dr Redder, 

We are pleased to inform you that your manuscript entitled "Critical factors for precise and efficient RNA cleavage by RNase Y in *Staphylococcus aureus*" has been formally accepted for publication in PLOS Genetics! Your manuscript is now with our production department and you will be notified of the publication date in due course.

With kind regards,

Anita Estes

PLOS Genetics

On behalf of:
